# Optimization of energy state transition trajectory supports the development of executive function during youth

Zaixu Cui[1], Jennifer Stiso[2], Graham L Baum[1], Jason Z Kim[2], David R Roalf[1], Richard F Betzel[3], Shi Gu[4], Zhixin Lu[2], Cedric H Xia[1], Xiaosong He[2], Rastko Ciric[1], Desmond J Oathes[1], Tyler M Moore[1], Russell T Shinohara[5], Kosha Ruparel[1], Christos Davatzikos[2,6], Fabio Pasqualetti[7], Raquel E Gur[1], Ruben C Gur[1], Danielle S Bassett[2,6,8,9,10†], Theodore D Satterthwaite[1†*]

[1]Departments of Psychiatry, University of Pennsylvania, Philadelphia, United States; [2]Departments of Bioengineering, University of Pennsylvania, Philadelphia, United States; [3]Department of Psychological and Brain Sciences, Indiana University, Bloomington, United States; [4]Department of Computer Science, University of Electronic Science and Technology, Chengdu, China; [5]Departments of Biostatistics, Epidemiology and Informatics, University of Pennsylvania, Philadelphia, United States; [6]Departments of Electrical and Systems Engineering, University of Pennsylvania, Philadelphia, United States; [7]Department of Mechanical Engineering, University of California, Riverside, United States; [8]Departments of Physics and Astronomy and Neurology, University of Pennsylvania, Philadelphia, United States; [9]Departments of Neurology, University of Pennsylvania, Philadelphia, United States; [10]Santa Fe Institute, Santa Fe, United States

**\*For correspondence:**
sattertt@pennmedicine.upenn.edu

[†]These authors contributed equally to this work

**Abstract** Executive function develops during adolescence, yet it remains unknown how structural brain networks mature to facilitate activation of the fronto-parietal system, which is critical for executive function. In a sample of 946 human youths (ages 8-23y) who completed diffusion imaging, we capitalized upon recent advances in linear dynamical network control theory to calculate the energetic cost necessary to activate the fronto-parietal system through the control of multiple brain regions given existing structural network topology. We found that the energy required to activate the fronto-parietal system declined with development, and the pattern of regional energetic cost predicts unseen individuals' brain maturity. Finally, energetic requirements of the cingulate cortex were negatively correlated with executive performance, and partially mediated the development of executive performance with age. Our results reveal a mechanism by which structural networks develop during adolescence to reduce the theoretical energetic costs of transitions to activation states necessary for executive function.

## Introduction

Executive function is essential for a wide range of cognitive tasks, and is strongly associated with both overall intelligence (*Arffa, 2007*) and academic performance (*Best et al., 2011*). Executive function undergoes protracted maturation during adolescence (*Best and Miller, 2010*; *Gur et al., 2012*), and its development is linked to the expansion of the cognitive and behavioral repertoire. Notably, executive deficits are linked to both increased morbidity associated with risk-taking behaviors (*Romer et al., 2009*) as well as a wide range of neuropsychiatric disorders (*Shanmugan et al.,*

**eLife digest** Adolescents are known for taking risks, from driving too fast to experimenting with drugs and alcohol. Such behaviors tend to decrease as individuals move into adulthood. Most people in their mid-twenties have greater self-control than they did as teenagers. They are also often better at planning, sustaining attention, and inhibiting impulsive behaviors. These skills, which are known as executive functions, develop over the course of adolescence.

Executive functions rely upon a series of brain regions distributed across the frontal lobe and the lobe that sits just behind it, the parietal lobe. Fiber tracts connect these regions to form a fronto-parietal network. These fiber tracts are also referred to as white matter due to the whitish fatty material that surrounds and insulates them.

Cui et al. now show that changes in white matter networks have implications for teen behavior. Almost 950 healthy young people aged between 8 and 23 years underwent a type of brain scan called diffusion-weighted imaging that visualizes white matter. The scans revealed that white matter networks in the frontal and parietal lobes mature over adolescence. This makes it easier for individuals to activate their fronto-parietal networks by decreasing the amount of energy required. Cui et al. show that a computer model can predict the maturity of a person's brain based on the energy needed to activate their fronto-parietal networks.

These changes help explain why executive functions improve during adolescence. This in turn explains why behaviors such as risk-taking tend to decrease with age. That said, adults with various psychiatric disorders, such as ADHD and psychosis, often show impaired executive functions. In the future, it may be possible to reduce these impairments by applying magnetic fields to the scalp to reduce the activity of specific brain regions. The techniques used in the current study could help reveal which brain regions to target with this approach.

2016), such as attention deficit hyperactivity disorder (ADHD) and psychosis (*Barkley, 1997*; *Wolf et al., 2015*).

Prior studies have consistently established that executive function relies on activity in a distributed network of fronto-parietal regions, including the dorsolateral prefrontal cortex, cingulate cortex, superior parietal cortex, and frontopolar cortex (*Alvarez and Emory, 2006*; *Mansouri et al., 2017*; *Niendam et al., 2012*; *Rottschy et al., 2012*; *Satterthwaite et al., 2013*). Notably, both functional (*Fair et al., 2007*; *Grayson and Fair, 2017*; *Gu et al., 2015*; *Power et al., 2010*) and structural (*Baum et al., 2017*; *Hagmann et al., 2010*; *Huang et al., 2015*) connectivity among these regions undergoes active remodeling during adolescence, with increased connectivity among executive regions, and diminished connectivity between executive regions and other systems such as the default mode network. As structural white matter networks are known to constrain both intrinsic connectivity and patterns of task-related activation (*Hermundstad et al., 2013*; *Honey et al., 2009*), it is possible that white matter networks develop during adolescence to facilitate dynamic transitions to fronto-parietal system activation states with lower theoretical energetic cost. However, research that seeks to relate developing white matter networks to the functional dynamics of the fronto-parietal executive system remains sparse.

Network control theory provides a powerful framework to address this gap in our knowledge. Previous work has shown that the brain becomes more theoretically controllable to all possible brain states (on average) through the control of individual brain regions during adolescence (*Tang et al., 2017*). Network control theory has the potential to provide novel insights regarding mechanisms needed to transition to executive states, as executive function exerts top-down control on other brain systems in a manner akin to control points in a dynamic network (*Gu et al., 2015*; *Tang et al., 2017*). Capitalizing on recent developments in network control theory (*Betzel et al., 2016*; *Gu et al., 2017*; *Stiso et al., 2019*), here we examine how the developing brain structural network supports the transition to a specific state necessary for executive function through the distributed control of multiple brain regions. Specifically, this new framework allows one to integrate information regarding network topology and patterns of brain activation within one mathematical model, in order to specify how theoretical neural dynamics are constrained by the structural connectome (*Tang and Bassett, 2018*). Such models assume that the activation state of the brain at a given time

is a linear function of the previous state, the underlying white matter network, and any additional control energy injected into the system (*Betzel et al., 2016*; *Gu et al., 2017*). From this paradigm, one can calculate the optimal energy cost to move the brain from one state to another given a structural network topology (*Betzel et al., 2016*; *Gu et al., 2017*; *Kim et al., 2018*). In the present work, we apply this new technique to a large sample of youth. Specifically, we investigated how the energetic cost of transitions to a fronto-parietal system activation state necessary for the executive function changes in response to the maturation of structural brain network. We hypothesized that maturation of structural brain networks would allow for the target activation state of the fronto-parietal executive system to be reached at a lower energetic cost.

To test this hypothesis, we capitalized on a large sample of youth (8–23 years) who completed neuroimaging as part of the Philadelphia Neurodevelopmental Cohort (PNC) (*Satterthwaite et al., 2014*). We examined how white matter networks (estimated using diffusion imaging) support the transition to a fronto-parietal system activation state. As described below, we demonstrate that the energy required to reach this state declines with age, especially within the fronto-parietal control network. Furthermore, we find that the whole-brain control energy pattern contains sufficient information to predict individuals' brain maturity across development. Finally, participants with better performance on executive tasks require less energetic cost in the bilateral cingulate cortex to reach this activation target, and the energetic cost of this region mediates the development of executive performance with age. Notably, these results could not be explained by individual differences in general network control properties, and were not present in alternative activation target states. Together, these results suggest that structural brain networks become optimized in development to minimize the energetic costs of transitions to activation states necessary for executive function through the distributed control of multiple brain regions.

## Results

### Network topology constrains the transition to a fronto-parietal activation state

In this study, we included 946 youths aged 8–23 years who were imaged as part of the PNC (*Figure 1—figure supplement 1*). Structural white matter networks were reconstructed for each participant from diffusion imaging data using probabilistic tractography and a standard parcellation of 232 regions. Capitalizing on recent advances in network control theory, we modeled how structural networks facilitate state transitions from an initial baseline state to the target state. In the initial state, all regions had an activity magnitude of 0. In the target state, regions in the fronto-parietal system had activity magnitude of 1, with all other regions having an activity magnitude of 0. Specifically, we defined the *trajectory* of a neural system to be the temporal path that the system traverses through diverse states, where the item *state* was defined as the vector of neurophysiological activity across brain regions at a single time point. Based on each participant's unique network topology, we estimated the regional energetic cost required for the brain to transition from the baseline to the fronto-parietal activation target state (*Betzel et al., 2016*; *Gu et al., 2017*; *Stiso et al., 2019*; *Figure 1—figure supplement 2* and *Figure 1a*). Formally, this estimation was operationalized as a multi-point network control optimization problem, where we aimed to identify the optimal trajectory between baseline and the fronto-parietal activation target state that minimizes both the energetic cost and the distance between the final state and the target state.

Results of this linear dynamical model indicate that the trajectory distance (i.e., the distance between current and target states) decreases with time until the desired target state is reached (*Figure 1—figure supplement 3a and b*). For each network node, we calculated the control energy cost, which provides an indication of where energy must be injected into the network to achieve the transition to the target state. Consistent with a recent methodological study (*Karrer et al., 2019*), and several recent empirical studies (*Betzel et al., 2016*; *Gu et al., 2017*; *Stiso et al., 2019*), the trajectory distance (*Figure 1—figure supplement 3b*) was inversely related to the time-dependent energy cost within subject (*Figure 1—figure supplement 3c*). We calculated the trajectory distance at each time point, which was defined as the Euclidean distance between the current brain state and the target brain state. A small distance suggests that the current vector of brain activity is similar to the target vector of brain activity. Across all subjects, we found the total trajectory distance of all

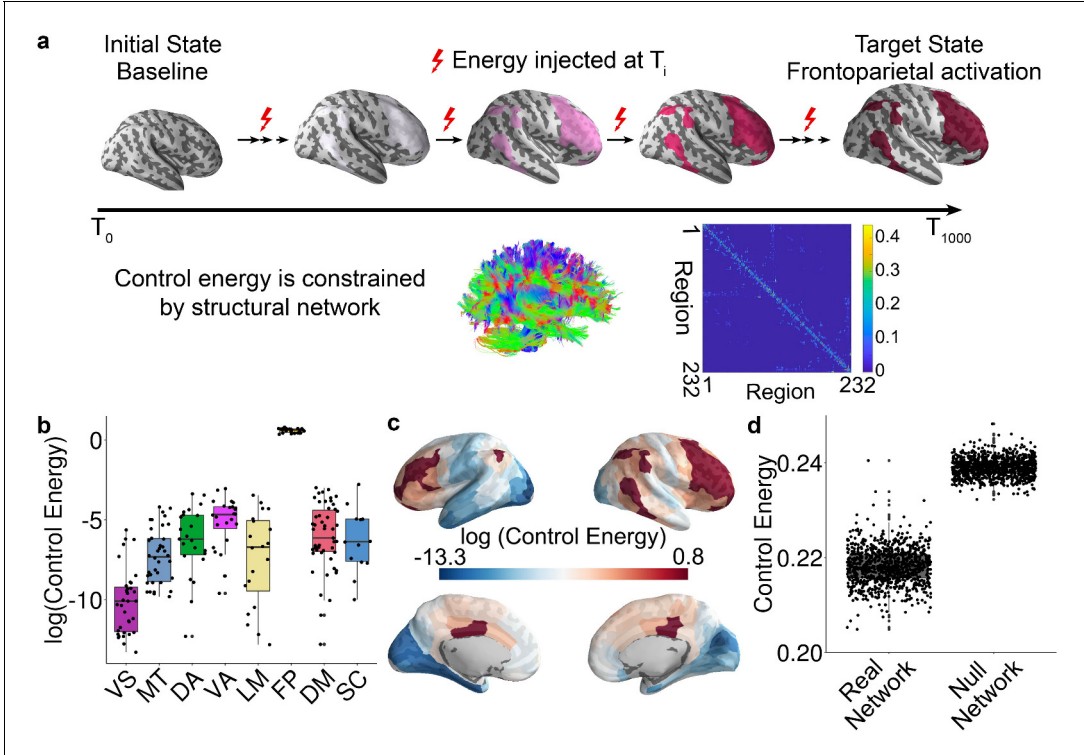

**Figure 1.** Schematic of the network control approach and the estimation of control energy. (a) From a baseline state, we calculated the control energy required to reach a fronto-parietal activation target state. This transition was calculated for each subject based on their structural brain network, which was estimated using diffusion imaging and probabilistic tractography. (b) The average energetic costs to reach the fronto-parietal activation target state varied by cognitive system, with the largest energetic costs being present in the fronto-parietal control network and the ventral attention network. (c) The regional control energy required to reach the fronto-parietal activation target. (d) The control energy cost of a transition to the fronto-parietal activation target state was significantly lower in real brain networks than in null model networks where the strength and degree distribution were preserved.

The online version of this article includes the following figure supplement(s) for figure 1:

**Figure supplement 1.** Sample construction.
**Figure supplement 2.** Functional brain networks defined by *Yeo et al. (2011)*.
**Figure supplement 3.** Relationship between trajectory distance and control energy.

time points was positively correlated with total control energy of all time points ($r = 0.97$, $p<2 \times 10^{-16}$, *Figure 1—figure supplement 3d*), suggesting that subjects whose state transition trajectory is long require more energy input to reach the target state. Prior literature has demonstrated that control energy cost is lower in human brain than in the brains of *Drosophila* and mouse to support diverse network dynamics (*Kim et al., 2018*), is related to network topology (*Betzel et al., 2016*; *Kim et al., 2018*) and reflects the magnitude of focal electrocorticography stimulation required to drive the brain to a target memory state in patients with medically refractory epilepsy (*Stiso et al., 2019*). Accordingly, here we used the control energy as a metric to summarize the optimal trajectory. We calculated the mean control energy of each system; the highest control energy was observed in systems involved in executive function (*Figure 1b and c*), including the fronto-parietal and ventral attention/cingulo-opercular systems (see *Figure 1—figure supplement 2*; *Yeo et al., 2011*).

Based on recent evidence that network control properties depend appreciably on the topological structure of the network (*Kim et al., 2018*; *Wu-Yan et al., 2017*), we next sought to demonstrate that the topological structure of brain networks facilitates this transition. We therefore compared the energetic cost of this transition in empirical brain networks to the energetic cost observed in null model networks. Specifically, we randomly permuted (100 times per participant) the placement of edge weights, while preserving the network degree and strength distribution. The mean whole brain

energetic cost of the null networks was significantly higher (p<$2 \times 10^{-16}$) than that of the empirical networks (*Figure 1d*), indicating that structural brain networks are topologically optimized to reduce the energetic costs of the transition to a fronto-parietal activation state.

## Energetic costs of the transition to a fronto-parietal activation state decline with development

Having shown that the topology of structural brain networks facilitates transitions to a fronto-parietal activation state, we next investigated how the energetic costs of this transition evolve in youth. We hypothesized that the energy required to make this transition would decline as networks were remodeled in development. Prior studies have demonstrated that the developmental changes of both brain structure and function could be either linear (*Hagmann et al., 2010*; *Wierenga et al., 2016*) or non-linear (*Grayson and Fair, 2017*; *Mills et al., 2016*; *Vandekar et al., 2015*). Therefore, we used generalized additive models (GAM) with penalized splines, which allowed us to rigorously characterize both linear and nonlinear effects while avoiding over-fitting. Age associations with control energy were examined at multiple scales, including the level of the whole brain, cognitive systems, and individual nodes. For all analyses, we included sex, handedness, in-scanner head motion, total brain volume, and total network strength as covariates. These analyses revealed that the whole-brain average energetic cost of the transition to the fronto-parietal activation state declined with age (Z = $-5.12$, p=$3.06 \times 10^{-7}$, Partial r = $-0.17$, 95% confidence interval (CI) = [$-0.23$,$-0.10$]; *Figure 2a*). Notably, analyses of cognitive systems indicated that age effects were heterogeneously distributed (*Figure 2b*), with the largest declines in control energy occurring in fronto-parietal (Z = $-5.30$, $P_{FDR}$ = $4.54 \times 10^{-7}$, Partial r = $-0.17$, CI = [$-0.23$,$-0.11$]; *Figure 2c*), visual (Z = $-4.25$, $P_{FDR}$ = $5.71 \times 10^{-5}$, Partial r = $-0.14$, CI = [$-0.20$,$-0.08$]), and motor (Z = $-3.20$, $P_{FDR}$ = $2.70 \times 10^{-3}$, Partial r = $-0.09$, CI = [$-0.15$,$-0.03$]) systems. In contrast, energetic costs within the limbic (Z = 8.69, $P_{FDR}$ <$2 \times 10^{-16}$, Partial r = 0.29, CI = [0.23, 0.35]) and default mode (Z = 2.86, $P_{FDR}$ = $5.66 \times 10^{-3}$, Partial r = 0.10, CI = [0.04, 0.17]) systems increased with age (see *Figure 2—figure supplement 1*). These system-level results aligned with analyses of individual network nodes; we found that the control energy of 49 regions decreased significantly with age ($P_{FDR}$ <0.05), including regions in the fronto-parietal control, visual, and motor systems. Furthermore, the control energy significantly increased with development in 30 regions ($P_{FDR}$ <0.05), which were mainly situated in limbic and default mode systems (*Figure 2d*).

Having found associations between age and control energy, we next conducted a series of eight additional analyses. First, we found that our results held true for a range of baseline initial states and a range of fronto-parietal activation target states. When 100 different initial baseline states were evaluated, we found that in all cases both the whole brain and the fronto-parietal system showed a significant decline in control energy with age (*Figure 2—figure supplement 2a*). Similarly, when 100 different target states of fronto-parietal activation were evaluated, we found that in all cases both the whole-brain and the fronto-parietal system showed a significant decline in control energy with age (*Figure 2—figure supplement 2b*).

Second, we evaluated whether age effects could be due to non-topological network properties by evaluating the presence of age effects in null networks where degree and strength distributions were preserved. We found that the significance level of age effects in null networks were smaller than those observed in the real network (p<0.01, 100 permutations), suggesting that the empirically measured developmental effects were indeed driven by changes in the network topology (*Figure 2—figure supplement 2c*). Third, it should be noted that we only constrained the state of regions in the fronto-parietal system. Therefore, the distance travelled by these off-target regions outside the fronto-parietal system were not included in our cost function for calculating optimal control energy. This choice also serves to ensure that our calculation of control energy is largely robust to both the initial and target states of other regions. To demonstrate the robustness of our results to our definition of the matrix **S**, we calculated the control energy cost using the same initial and target states as in the main analyses but constraining the whole brain. Results showed that there is a high correlation (r = 0.94, p<$2 \times 10^{-16}$) between the whole-brain control energy cost when constraining the whole brain and that when constraining the fronto-parietal system only (*Figure 2—figure supplement 2d*).

Fourth, we assessed whether the structural network optimized the transition to an a priori motor system activation target (*Figure 1—figure supplement 2*; *Yeo et al., 2011*). Results indicated that the mean whole brain energetic cost of the null networks was significantly higher (p<$2 \times 10^{-16}$) than

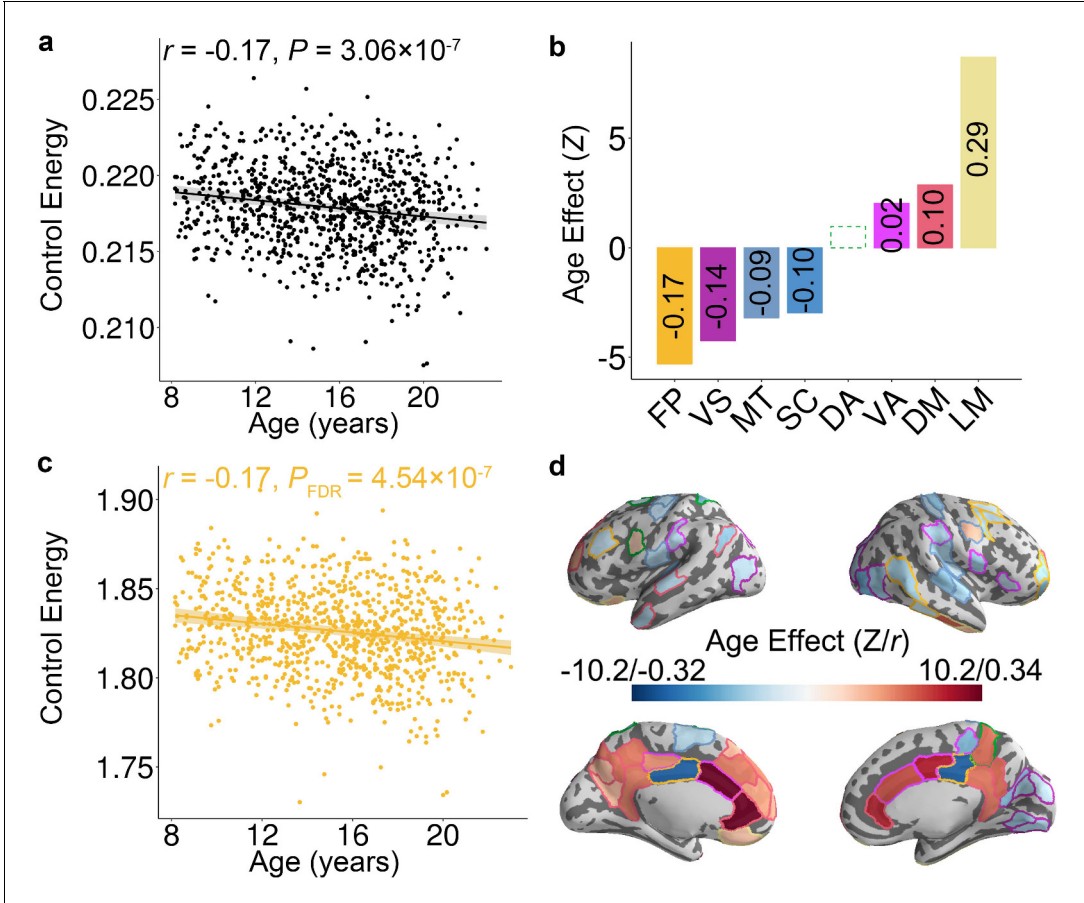

**Figure 2.** Control energy evolves with age in youth. (a) The mean whole-brain control energy required to reach the fronto-parietal activation target declines with age. (b) Control energy declines significantly with age in the fronto-parietal, visual, motor and subcortical systems. In contrast, control energy increased in the ventral attention, default mode and limbic systems. For each system with a significant association, the effect size is reported (in each bar) as the partial correlation between system-level control energy and age while controlling for the covariates. There is one outlier in the scatter plot of ventral attention system (*Figure 2—figure supplement 1c*) and the age-related changes of control energy was not significant (p=0.11) in this system after removing the outlier. (c) The control energy of the fronto-parietal system declines significantly with age. (d) The age effect of control energy for each brain region. The color of the contour of each brain region represents the cognitive system for each region (see *Figure 1—figure supplement 2*). In the scatterplots shown in panels (a and c), data points represent each subject (n = 946), the bold line indicates the best fit from a general additive model, and the shaded envelope denotes the 95% confidence interval. It should be noted that Z value was derived from the general additive model, which captures both linear and nonlinear relationships; the partial correlation reflects only linear relationships. VS: visual; MT: motor; DA: dorsal attention; VA: ventral attention; LM: limbic; FP: fronto-parietal; DM: default mode; SC: subcortical.

The online version of this article includes the following figure supplement(s) for figure 2:

**Figure supplement 1.** Scatter plots of significant age effects of control energy at the system scale.

**Figure supplement 2.** Specificity and sensitivity analyses provide convergent results.

**Figure supplement 3.** Age effects at the whole brain, cognitive system, and nodal levels remain after controlling for the (a) modal controllability and (b) network modularity.

**Figure supplement 4.** Convergent results from a target state defined by a working memory task that recruits the fronto-parietal system.

that of the empirical networks (*Figure 2—figure supplement 2e* left), suggesting that the lower energetic cost of activtivating thefronto-parietal system was not unique, but was present when activating other systems as well. We further evaluated the age effects of control energy cost to activate the motor system. As the age range of 8–23 years is a critical period in the development of executive function rather than motor function, we expected weaker age effects in the motor system. We found that the whole-brain control energy required to transition to the motor system activation did not significantly change over the age range studied (Z = 1.48, p=0.14, Partial $r$ = 0.05, CI = [−0.02, 0.11]; *Figure 2—figure supplement 2e* right).

Fifth, we evaluated whether our developmental results could be explained by *modal controllability*. Modal controllability reflects the extent to which all dynamic modes of a system will change in response to small changes at a single node (*Gu et al., 2015*). If an individual has high modal controllability, it suggests that the underlying brain structural network was optimized to support efficient state transitions to diverse states. In line with this intuition, modal controllability increases with development in youth as flexible switching between patterns of brain activity becomes more common (*Tang et al., 2017*). Controlling for modal controllability did not alter our results (*Figure 2—figure supplement 3a*). Specifically, while controlling for modal controllability, average control energy of the whole-brain and fronto-parietal system both significantly declined with age (whole-brain: Z = −6.00, p=2.09 × 10$^{-9}$, Partial *r* = −0.19, CI = [−0.25,–0.13]; fronto-parietal: Z = −9.95, $P_{FDR}$ <2 × 10$^{-16}$, Partial *r* = −0.32, CI = [−0.37,–0.26]).

Sixth, because the modularity of brain networks evolves with age, one could ask whether that evolution impacts the observed assocations with control energy (*Baum et al., 2017*; *Hagmann et al., 2010*; *Huang et al., 2015*). However, we found that results remained consistent after controlling for network modularity in all analyses (*Figure 2—figure supplement 3b*). For example, average control energy of the whole brain and of the fronto-parietal system both significantly declined with age after controlling for network modularity (whole-brain: Z = −3.95, p=7.73 × 10$^{-5}$, Partial *r* = −0.13, CI = [−0.19,–0.07]; fronto-parietal: Z = −4.31, $P_{FDR}$ = 6.46 × 10$^{-5}$, Partial *r* = −0.14, CI = [−0.20,–0.08]). We further assessed whether the increasing segregation of fronto-parietal system during youth (*Baum et al., 2017*) could explain the age effect of control energy. Results remained consistent after controlling for the average participation coefficient within the fronto-parietal system when examining age-related differences in the average control energy of the fronto-parietal system (Z = −4.64, p=3.51 × 10$^{-6}$, Partial *r* = −0.15, CI = [−0.21,–0.09]).

Seventh, we assessed whether connectivity within the fronto-parietal system or between the fronto-parietal and other systems could explain observed associations between age and control energy. Specifically, we calculated the sum of all the connections within fronto-parietal system and also the sum of all the connections between the fronto-parietal system and other systems. While controlling for within fronto-parietal connectivity strength, the control energy in the fronto-parietal system still significantly declined with development (Z = −3.53, p=0.0004, Partial *r* = −0.12, CI = [−0.18,–0.06]). Similarly, while controlling for the connectivity strength between the fronto-parietal system and other systems, the control energy in the fronto-parietal system still significantly declined with development (Z = −4.88, p=1.06 × 10$^{-6}$, Partial *r* = −0.16, CI = [−0.22,–0.10]).

Finally, in our main analyses, we specified the target state as regions within the fronto-parietal system, with each region having a magnitude of 1. As a final step, we also considered a biologically recorded target state defined as the average activation pattern elicited by an *n*-back working memory task that reliably recruits the fronto-parietal system (*Figure 2—figure supplement 4a*). Using this alternative target state, we found that the control energy cost of the real network was significantly lower than null networks (*Figure 2—figure supplement 4b*). As in the main analyses, the control energy cost was highest in the fronto-parietal system (*Figure 2—figure supplement 4c*). Similarly, the whole-brain average control energy cost (Z = −7.59, p=3.26 × 10$^{-14}$, Partial *r* = −0.25, CI = [−0.30,–0.18]; *Figure 2—figure supplement 4d*) and average control energy in the fronto-parietal system (Z = −5.26, $P_{FDR}$ = 2.92 × 10$^{-7}$, Partial *r* = −0.17, CI = [−0.23,–0.11]; *Figure 2—figure supplement 4e*) both significantly declined with age. Nodal analyses provided convergent results, revealing that the control energy in nodes within the fronto-parietal system significantly declined with age (*Figure 2—figure supplement 4f*).

## Patterns of control energy can predict brain maturity

Having established that the control energy required to reach the fronto-parietal activation state changes with age on a regional and system-level basis using mass-univariate analysis, we next evaluated the developmental changes of control energy using multivariate pattern analysis. Multivariate pattern analysis complements mass-univariate analysis, as mass-univariate analysis investigates each feature (i.e., control energy of one brain region) in isolation. In contrast, multivariate pattern analyses are sensitive to the spatially distributed pattern of features (*Davatzikos, 2004*; *Haynes, 2015*; *Haynes and Rees, 2006*; *Norman et al., 2006*). To provide an integrated view of this high-dimensional data, we used multivariate pattern analysis to determine whether spatially distributed patterns of control energy could accurately predict participant age. Specifically, we applied ridge regression

with nested two-fold cross validation (2F-CV, see *Figure 3—figure supplement 1*) to identify an individual participant's age in an unbiased fashion using the multivariate pattern of regional control energy. Specifically, we divided all subjects into two subsets based on age, with the first subset used as a training set and the second subset used as a testing set. Within the training set, we used inner 2F-CV to select an optimal regularization parameter ($\lambda$). Then, we trained a model using the training data and predicted the brain maturity (i.e., 'brain age') of participants in the testing set (*Dosenbach et al., 2010*; *Franke et al., 2010*). The significance of the model was evaluated using permutation testing, where the correspondence between a subject's control energy features and their age was permuted at random. This analysis revealed that the multivariate pattern of control energy could predict an unseen individual's age (*Figure 3a* and *Figure 3—figure supplement 2a and b*): the correlation between the predicted 'brain age' and chronological age was 0.63 ($p < 0.001$) after controlling for the covariates, and the mean absolute error (MAE) was 2.16 years ($p < 0.001$). For completeness, we also repeated this procedure while reversing the training and test sets, which yielded very similar results (partial $r = 0.58$, $p < 0.001$; MAE = 2.27, $p < 0.001$; *Figure 3a* and *Figure 3—figure supplement 2c and d*). We further examined model weights at the level of individual network nodes. The regions that contributed the most to the prediction of brain maturity aligned with mass-univariate analyses, and included the dorsolateral and ventrolateral prefrontal cortex, the cingulate cortex, superior parietal cortex, and lateral temporal cortex (*Figure 3b*). In order to ensure that our initial split of the data was representative, we repeated this analysis with 100 random splits, which returned highly consistent results (mean partial $r = 0.61$, mean MAE = 2.21 years).

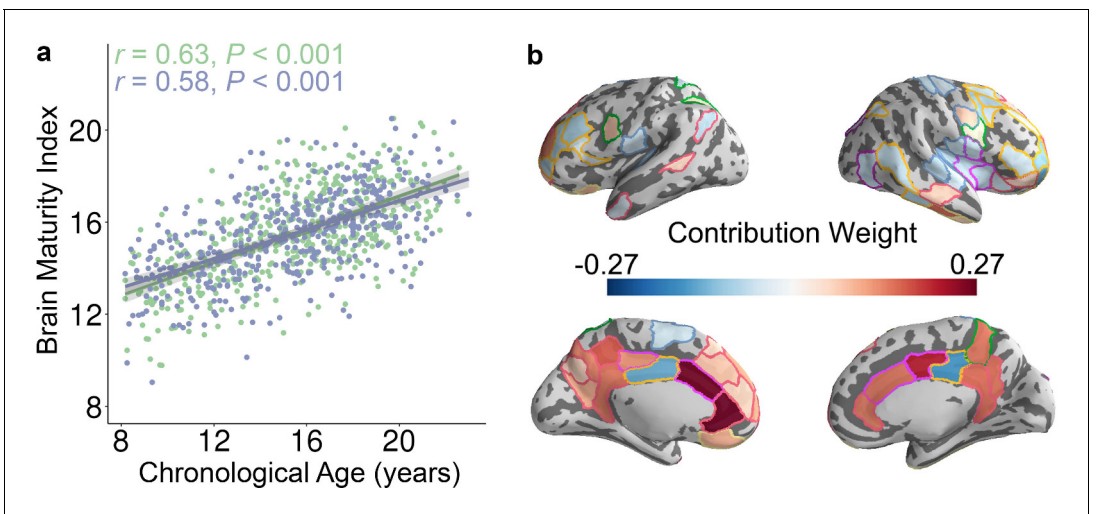

**Figure 3.** The whole-brain control energy pattern contains sufficient information to predict brain maturity in unseen individuals. (a) The predicted brain maturity index was significantly related to the chronological age in a multivariate ridge regression model that used 2-fold cross validation (2F-CV) with nested parameter tuning. The complete sample of of subjects was divided into two subsets according to age rank. The blue color represents the best-fit line between the actual score of the first subset of subjects and their scores predicted by the model trained using the second subset of subjects. The green color represents the best-fit line between the actual score of the second subset of subjects and their scores predicted by the model trained using the first subset of subjects. (b) Regions with the highest contribution to the multivariate model aligned with mass-univariate analyses and included frontal, parietal, and temporal regions. We displayed the 79 regions with the highest contribution, to facilitate comparisons with mass-univariate analyses (where there were 79 regions with significant age effects).

The online version of this article includes the following figure supplement(s) for figure 3:

**Figure supplement 1.** Schematic overview of one outer loop of the nested 2-fold cross-validation (2F-CV) prediction framework.

**Figure supplement 2.** The histograms of the permutation distribution of the (a) correlation $r$ and (b) MAE with the first subset used as a training set and the second subset used as a testing set, and (c) correlation $r$ and (d) MAE with the first subset used as the testing set and the second subset used as training set.

## Participants with higher executive function need less energy to activate the fronto-parietal network

Lastly, we investigated the cognitive implications of individual differences in control energy. Specifically, we expected that participants with higher executive performance on a standardized cognitive battery would require reduced control energy to activate the fronto-parietal system. In order to ensure that associations were present above and beyond the observed developmental effects, we controlled for linear and nonlinear effects of age in addition to the other covariates described above. While we did not find effects at the whole-brain or systems level, two regions survived after FDR correction at nodal level. Specifically, reduced control energy within two regions in the fronto-parietal control system – the left and right middle cingulate cortex – was associated with higher executive function (Left: Z = −3.65, $P_{FDR}$ = 0.032, Partial $r$ = −0.13, CI = [−0.19–0.06]; Right: Z = −4.49, $P_{FDR}$ = 0.002, Partial $r$ = −0.15, CI = [−0.21–0.08]; *Figure 4a and b*).

Given that control energy reflects the topology of diffusion network (*Kim et al., 2018*) and prior study showed that diffusion network properties mediated the age-related development of executive function (*Baum et al., 2017*), we conducted mediation analyses to investigate the extent to which control energy accounted for the association between age and executive function. Using a bootstrapped mediation analysis while adjusting for the covariates described above (See Materials and methods), we found that control energy in both the left (β = 0.03, p=0.001, 95% confidence interval = [0.01, 0.04]; *Figure 4c*) and right middle cingulate cortex (β = 0.03, p<0.001, 95%

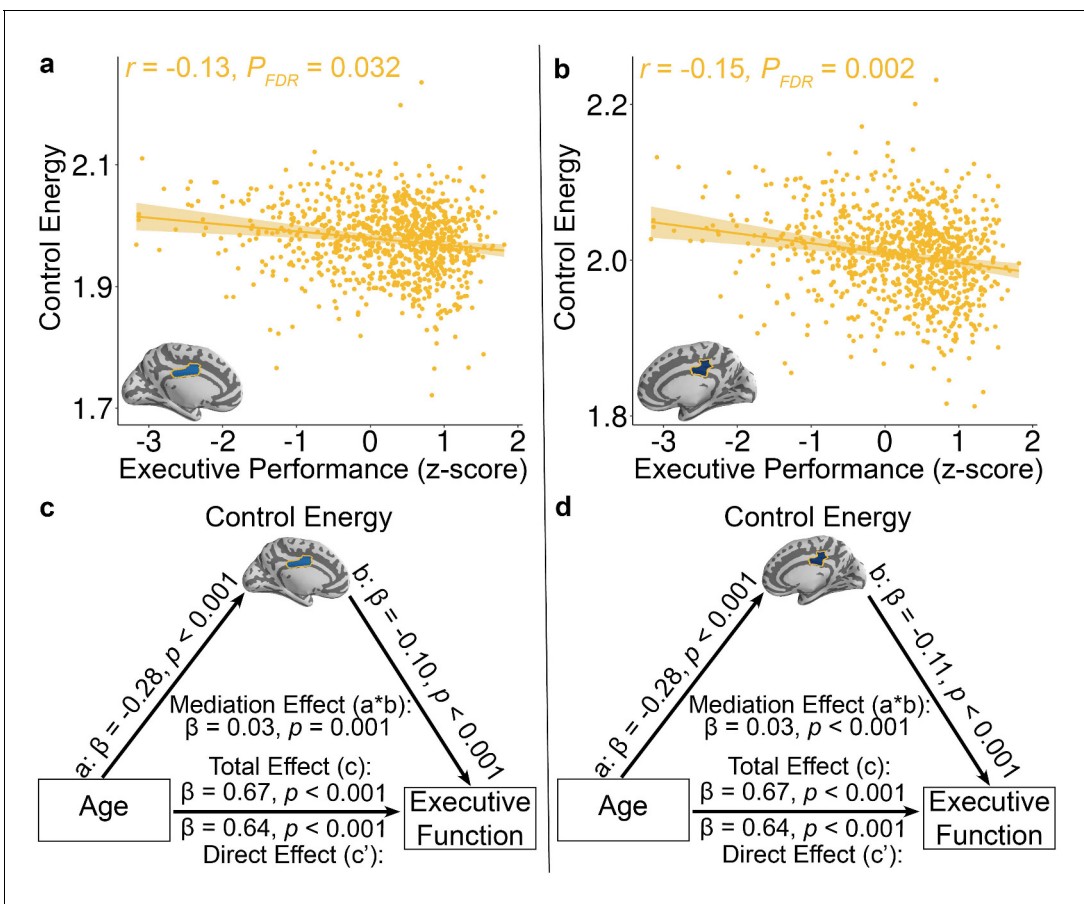

**Figure 4.** Reduced control energy in both the **a**, left and **b**, right mid-cingulate cortex was associated with higher executive performance. Of all brain regions examined, only the left and right mid-cingulate cortex survived FDR correction. Data points represent each subject (n = 944), the bold line indicates the best linear fit, and the shaded envelope denotes the 95% confidence interval.The yellow color indicates that the two regions belong to the fronto-parietal system (*Figure 1—figure supplement 2*). The control energy of both (**c**) left and (**d**) right mid-cingulate cortex partially mediates the improvement of executive function with age. Significance of mediation effect was assessed using bootstrapped confidence intervals.

confidence interval = [0.02, 0.05]; *Figure 4d*) mediated the development of executive function with age.

## Discussion

Using a large sample of youths (8–23 years) and a generative model of brain network function, we demonstrated that the control energy theoretically required to transition to a fronto-parietal activation state declines with age in youth. Furthermore, the multivariate pattern of the whole-brain control energy predicted the brain maturity of unseen individual participants. These results could not be explained by general network control property and were not observed in analyses stipulating alternative activation targets. Finally, we found that individuals who had higher executive function required lower control energy in the bilateral middle cingulate cortex to activate the fronto-parietal system, and the control energy of this region partially mediated the development of executive performance with age. These results suggest that maturation of structural brain networks may facilitate transitions to fronto-parietal activation states that support executive function.

While prior work has consistently demonstrated similarities in the configuration of structural and functional connectivity (*Hagmann et al., 2010*; *Honey et al., 2009*; *Mollink et al., 2019*), these studies did not evaluate how brain structural networks constrain functional dynamics (*Avena-Koenigsberger et al., 2017*). Recently, using network control methods, several studies have modeled the structural network and brain activation in one framework to describe how the structural connectome may theoretically constrain dynamic transitions between two activation states (*Betzel et al., 2016*; *Gu et al., 2017*; *Kim et al., 2018*; *Stiso et al., 2019*). By extending this framework in a large population, we found that the theoretical energetic cost of brain state transition from a baseline to a fronto-parietal activation state was lower in real brain networks compared to null networks that preserved basic properties such as degree and strength distribution. This result was consistent with prior studies showing that human structural brain networks exhibit non-random topological properties, such as both high clustering and segregated modules (*Bullmore and Sporns, 2009*; *Gong et al., 2009b*; *Hagmann et al., 2008*). This non-random topological organization could support the energetically efficient activation of functional systems (*Avena-Koenigsberger et al., 2017*), and we specifically demonstrate that it supports the efficient control of transitions to the precise activation states required for executive function.

Structural connectivity remodels during youth, with increasing segregation (*Huang et al., 2015*), and greater integration (*Hagmann et al., 2010*). Together, these impose a stronger constraint on functional dynamics (*Hagmann et al., 2010*). However, prior work had not explored how the development of structural connectivity supports the emergence of functional activation relevant to executive function. Our results indicate that the control energy theoretically required to transition to the fronto-parietal activation state declines with development, and suggest that the topological organization of structural connectivity supports more energetically efficient signal transformation to activate the fronto-parietal network. Consistent with our results, prior work has demonstrated that metabolism costs declined in youth at both resting-state (*Jog et al., 2016*; *Takahashi et al., 1999*) and during working memory tasks (*Jog et al., 2016*).

Examination of individual cognitive systems revealed that this decline in whole-brain energy was driven by reduced energetic costs within the fronto-parietal system. In particular, substantial negative associations between age and control energy were observed in lateral prefrontal cortex and middle cingulate cortex, which are responsible for preparation, execution, monitoring and switching of tasks (i.e., working memory, attention, inhibitory control, etc.) (*Alvarez and Emory, 2006*; *Apps et al., 2013*; *Niendam et al., 2012*; *Rottschy et al., 2012*). Such reduced regional energetic cost suggests that structural brain networks may mature to allow for neural events that occur in these regions to impact the broad activation state of the entire network more efficiently (*Baum et al., 2017*; *Hagmann et al., 2010*), and more easily drive the brain towards the fronto-parietal activation state associated with demanding executive tasks (*Satterthwaite et al., 2013*; *Thomason et al., 2009*).

In contrast, the energetic cost of regions within the limbic and default mode systems increased with age. This localization of costs suggests that these regions become less able to move the brain to a fronto-parietal activation state as development progresses. This result was consistent with previous studies using both structural and functional connectivity data, which have shown that the fronto-

parietal system becomes more segregated in development from other systems in association cortex (*Grayson and Fair, 2017*; *He et al., 2019*; *Huang et al., 2015*; *Lee and Telzer, 2016*; *Sherman et al., 2014*), including the default mode (*Sherman et al., 2014*) and limbic (*Lee and Telzer, 2016*) systems. This maturation may potentially allow for functional specialization and a reduction of interference.

It should be noted that in prior work we demonstrated that another network control property – modal controllability – increased with age (*Tang et al., 2017*). However, modal controllability quantifies the general controllability necessary to reach *all* possible states, and therefore is not sufficient to answer questions about *specific* patterns of activity. Given the importance of executive function in youth to academic achievement (*Best et al., 2011*), risk taking behaviors (*Romer et al., 2009*), and psychopathology (*Shanmugan et al., 2016*), here we sought to understand how structural networks develop to facilitate the transition to fronto-parietal activation states that are necessary for executive function. We therefore leveraged recent advances in network control theory that allow for examining the controllability of a specific biologically meaningful target state and using multi-point control – rather than examining the general controllability and using the single point control methods employed previously (*Tang et al., 2017*). We have previously demonstrated the validity of this new control framework using mathematics, animal data, brain stimulation data, and human brain imaging data (*Betzel et al., 2016*; *Kim et al., 2018*; *Stiso et al., 2019*). As the brain is constantly receiving input from multiple sensory modalities and top down cognitive processes, we believe that multi-point control is substantially more biologically plausible for understanding executive function than single-point control frameworks.

Using this framework, we found that the energetic costs to transition to a specific fronto-parietal activation target state decline with age, and models trained using control energy accurately encode brain maturity in unseen data. Importantly, we demonstrate that our current results are not simply a result of increasing single-point modal controllability: when modal controllability was included as a model covariate, our results remained unchanged. Critically, the increase of general controllability (i. e., modal controllability) with development does not provide information regarding control of the transition to a specific brain state transition. While the brain becomes more controllable on average as shown in *Tang et al. (2017)*, the control energy required to arrive a specific activation state could decrease, not change, or even possibly increase. Consistent with this intuition, we demonstrated that the control energy cost to activate the motor system did not significantly change during development. The result accords with prior evidence indicating that motor development precedes executive development, and is largely complete by late childhood or early adolescence (*Andersen, 2003*; *Gogtay et al., 2004*) and suggests that the observed developmental changes in control energy may be specific for transitions to activation states recruiting higher-order cognitive systems, which undergo protracted maturation.

Additionally, we found null networks that preserved degree and strength distribution did not represent similar developmental changes of control energy, suggesting our results were driven by topological structure of the brain network. Our results suggest that neither the overall modularity of the structural network nor the segregation of the fronto-parietal system, which both mature during youth (*Baum et al., 2017*), could explain the association between age and control energy. This result was consistent with our recent study showing that the modularity can be positively, non-monotonically, and non-significantly related to control energy depending on the architecture of the structural adjacency matrix (*Patankar et al., 2020*). We found that connectivity strength within the fronto-parietal system partially contributed to reduced energy requirements for activating the fronto-parietal areas in development. The energy required for a state transition depends upon a system's response to an energetic perturbation. The simplest such perturbation engenders an impulse response (*Karrer et al., 2020*). In *Srivastava et al., 2020*, we show that the impulse response of the system is formally related to the network communicability, suggesting that paths of all lengths – not just direct connections – contribute to the control energy. However, the precise relationship between network topology and control energy is unknown. In prior theoretical work, we made some progress in linking the underlying graph architecture to the observed control energy (*Kim et al., 2018*). Specifically, given we have the ability to control a subset of nodes in the network, we showed that the more strongly and more diversely the control nodes are connected to the non-control nodes, the less energy is required to control the network on average. We provided an analytical derivations of the expressions relating a network's minimum control energy to its connectivity between control and

non-control nodes, as well as offering an intuitive geometric representation to visualize this relationship, and rules for modifying edges to alter control energy in a predictable manner (*Kim et al., 2018*). However, this theory is not applicable to our current study because the theory requires that only a set of brain regions be used for control, whereas here we set the control set to be the entire brain. More formal theoretical work is necessary to identify the topological properties that explain control energy.

Our main results regarding brain development (as well as the supplementary analyses described above) used a mass-univariate analysis approach, where the association between the control energy of each region was modeled separately. Complementary analyses sought to identify distributed multivariate patterns of control energy, which could be used to predict the brain maturity of unseen individuals. Such an approach is similar to prior studies that have used structural (*Franke et al., 2010*), functional (*Dosenbach et al., 2010*), or diffusion (*Erus et al., 2015*) based imaging to predict brain development. Here, we used a rigorous split half validation framework with nested parameter tuning. We found that the complex pattern of control energy could be used to predict individual brain maturity. The feature weights from this multivariate model were generally consistent with findings from mass-univariate analyses, underscoring the robustness of these results to the methodological approach. In this context, control energy could have potential to determine whether individuals display either precocity or delay in specific dynamic aspects of brain maturation, which may be relevant to studying developmental disorders and neuropsychiatric syndromes (*Dosenbach et al., 2010*; *Erus et al., 2015*).

Furthermore, while controlling for age, we observed a significant negative correlation between control energy of both the bilateral middle cingulate cortex and executive function performance. The middle cingulate is a component of the fronto-parietal control system (*Fair et al., 2007*; *Yeo et al., 2011*) and is critical for executive tasks such as performance monitoring, error detection, and task switching (*Apps et al., 2013*; *Vogt, 2016*). This result suggests that individuals with better executive function may be able to transition to the fronto-parietal activation state more easily. This result is consistent with prior studies showing that executive function training improves efficiency in activating the executive network (*Kozasa et al., 2012*; *Rueda et al., 2012*), and that subjects with higher cognitive performance have lower brain cerebral metabolism cost in resting-state (*Bastin et al., 2012*) and have improved metabolic control (*Ryan et al., 2006*). Moreover, the decline of control energy of the bilateral middle cingulate cortex partially mediated the observed improvement of executive function with age. This result is consistent with prior literature suggesting that the optimization of the structural network is associated with better executive function (*Baum et al., 2017*; *Wen et al., 2011*), as the decline of control energy cost reflects the optimization of the structural brain network (*Kim et al., 2018*). However, it should be noted that this mediation effect was small, as the direct effect was much larger than the indirect effect.

While brain network dynamics are known to be nonlinear, we used a linearization model here. It should be acknowledged that this linearization constrains the model's predictive power to short time-scales and to states in the immediate vicinity of the operating point. Nonetheless, it has been consistently demonstrated that linearization offers fundamental insights into nonlinear dynamics. For example, *Honey et al. (2009)* shows that predictions of function from structure can be obtained with both linear and nonlinear models. Second, if the linearized system is locally controllable along a specific trajectory in state space, then the original nonlinear system is also controllable along the same trajectory (*Coron, 2007*; *Yan et al., 2017*). Finally, linear controllers are often used to control nonlinear systems through gain scheduling in flight and process control (*Leith and Leithead, 2000*). Accordingly, while the linear control trajectories in our work suffer from consequences due to linearization, they also serve as a natural and informative prior for future work in principled neural stimulation and nonlinear control when more is known about the exact nature of the brain's nonlinearity.

Several limitations should be noted. First, all data presented here were cross-sectional, which precludes inference regarding within-individual developmental effects. Ongoing follow-up of the PNC will yield informative longitudinal data, as will other large-scale studies such as the Adolescent Brain and Cognitive Development Study. Second, it should be noted that probabilistic tractography methods remain limited in their ability to fully resolve the complex white matter architecture of the human brain. However, these methods are currently considered state-of-the-art, and may be superior to tensor-based tractography in resolving crossing fibers (*Behrens et al., 2007*). Third, it should be noted that motion artifact is a major potential confound for any study of brain development, and

prior studies by our group and others have shown that motion artifact can bias estimates of tractography and confound developmental inference (*Baum et al., 2018*). However, to limit the impact of this confound, we conducted rigorous quality assurance and included in-scanner motion as a covariate in all analyses. Fourth, it should be noted that most univariate effect sizes at system level reported in our work were small. However, prior work has consistently demonstrated that small samples systematically inflate the apparent effect size (*Yarkoni, 2009*), whereas large samples (such as this one, n = 946) provide a much more accurate estimate of the true effect size. Furthermore, in contrast to the small univariate effects observed at system level, some univariate effects at nodal level were at medium size and results from the multivariate analyses yielded large effect sizes in unseen data. Finally, the differences between the developmental effects with the motor system target state and that with the fronto-parietal system target state could be due to the differences in size or in spatial congruence between the two target states. However, the non-significant developmental effect with motor target state is consistent with the evidence that motor development is largely complete by late childhood or early adolescence (*Andersen, 2003*; *Gogtay et al., 2004*).

These potential limitations notwithstanding, we demonstrated that the topological structure of white matter networks is optimized during development to facilitate transitions to a fronto-parietal activation state. Moving forward, this framework may be useful for understanding the developmental substrates of executive dysfunction in diverse psychiatric disorders including psychosis and ADHD. Improved knowledge regarding both normal network development and abnormalities associated with psychopathology is a prerequisite for developing individualized interventions to alter disease trajectories and improve patient outcomes. In the future, advances in non-invasive neuromodulatory therapies may allow for targeted stimulation of specific brain regions that are optimally situated within the brain's control architecture to facilitate transitions to specific target states (*Medaglia et al., 2018*). Such advances could potentially aid in the treatment of the wide range of neuropsychiatric disorders marked by executive dysfunction (*Braun et al., 2018*).

## Materials and methods

### Participants

All subjects or their parent/guardian provided informed consent, and minors provided assent. The Institutional Review Boards of both Penn and CHOP approved study procedures. Overall, 1601 participants were enrolled (*Satterthwaite et al., 2014*). However, 340 subjects were excluded owing to clinical factors including medical disorders that could affect brain function, current use of psychoactive medications, prior inpatient psychiatric treatment, or an incidentally encountered structural brain abnormality. Among the 1261 subjects eligible for inclusion, 54 subjects were excluded for a low quality T1-weighted image or errors in the FreeSurfer reconstruction. Of the remaining 1207 subjects with a usable T1 image, 128 subjects were excluded because of the lack of a complete diffusion scan. Of the 1079 subjects with complete diffusion data, 110 subjects failed quality assurance as part a rigorous quality assurance protocol for diffusion MRI (*Roalf et al., 2016*). Additionally, 20 subjects were excluded because they had no field map for distortion correction. Finally, of the remaining 949 subjects, three subjects were excluded due to incomplete image coverage during brain parcellation, yielding a final sample of 946 participants (*Figure 1—figure supplement 1*).

### Cognitive assessment

The Penn computerized neurocognitive battery (Penn CNB) was administered to all participants during a separate session from neuroimaging. The CNB consists of 14 tests adapted from tasks applied in functional neuroimaging to evaluate a broad range of cognitive domains (*Gur et al., 2012*). These domains include executive control (abstraction and mental flexibility, attention, working memory), episodic memory (verbal, facial, spatial), complex cognition (verbal reasoning, nonverbal reasoning, spatial processing), social cognition (emotion identification, emotion differentiation, age differentiation) and sensorimotor and motor speed. Accuracy and speed for each test were z-transformed and summarized into an efficiency score. A factor analysis was used to summarize these efficiency scores into four factors (*Moore et al., 2015*), including executive function, complex reasoning, memory, and social cognition. Here, we focused on the executive function factor score. Of the sample of 946 participants with complete imaging data that passed quality assurance, two participants had

incomplete cognitive data. Accordingly, 944 participants were used in the analysis examining the association between cognition and control energy.

## Image acquisition

As previously described (*Satterthwaite et al., 2014*), all MRI scans were acquired on the same 3T Siemens Tim Trio whole-body scanner and 32-channel head coil at the Hospital of the University of Pennsylvania.

### Structural MRI

Prior to dMRI acquisitions, a 5 min magnetization-prepared, rapid acquisition gradient-echo T1-weighted (MPRAGE) image (TR = 1810 ms; TE = 3.51 ms; FOV = 180 × 240 mm$^2$, matrix = 192 × 256, effective voxel resolution = 0.9 × 0.9×1 mm$^3$) was acquired. This high-resolution structural image was used for tissue segmentation and parcellating gray matter into anatomically defined regions in native space.

### Diffusion MRI

Diffusion MRI scans were acquired using a twice-refocused spin-echo (TRSE) single-shot echo-planar imaging (EPI) sequence (TR = 8100 ms; TE = 82 ms; FOV = 240 mm$^2$/240 mm$^2$; Matrix = RL:128, AP:128; Slices: 70, in-plane resolution (x and y) 1.875 mm$^2$; slice thickness = 2 mm, gap = 0; flip angle = 90/180/180; volumes = 71; GRAPPA factor = 3; bandwidth = 2170 Hz/pixel; PE direction = AP). This sequence used a four-lobed diffusion encoding gradient scheme combined with a 90-180-180 spin-echo sequence designed to minimize eddy-current artifacts. For dMRI acquisition, a 64-direction set was divided into two independent 32-direction imaging runs in order to ensure that the scan duration was more tolerable for young subjects. Each 32-direction sub-set was chosen to be maximally independent such that they separately sampled the surface of a sphere (*Jones et al., 2002*). The complete sequence consisted of 64 diffusion-weighted directions with b = 1000 s/mm$^2$ and 7 interspersed scans where b = 0 s/mm$^2$. The total duration of dMRI scans was approximately 11 min. The imaging volume was prescribed in axial orientation covering the entire cerebrum with the topmost slice just superior to the apex of the brain (*Satterthwaite et al., 2014*).

### N-back task fMRI

Blood oxygen level-dependent fMRI was acquired using a whole-brain, single-shot, multislice, gradient-echo echoplanar sequence with the following parameters: 231 volumes; TR = 300 ms; TE = 32 ms; flip angle = 90; FOV = 192 × 192 mm; matrix = 64 × 64; 46 slices; slice thickness/gap = 3/0 mm; effective voxel resolution = 3 × 3×3 mm.

### Field map

In addition, a B0 field map was derived for application of distortion correction procedures, using following the double-echo, gradient-recalled echo (GRE) sequence: TR = 1000 ms; TE1 = 2.69 ms; TE2 = 5.27 ms; 44 slices; slice thickness/gap = 4/0 mm; FOV = 240 mm; effective voxel resolution = 3.8 × 3.8×4 mm.

### Scanning procedure

Before scanning, to acclimate subjects to the MRI environment, a mock scanning session where subjects practiced the task was conducted using a decommissioned MRI scanner and head coil. Mock scanning was accompanied by acoustic recordings of the noise produced by gradient coils for each scanning pulse sequence. During these sessions, feedback regarding head movement was provided using the MoTrack motion tracking system (Psychology Software Tools). Motion feedback was given only during the mock scanning session. To further minimize motion, before data acquisition, subjects' heads were stabilized in the head coil using one foam pad over each ear and a third over the top of the head.

## Image processing

### Structural image processing and network node definition

The structural image was processed using FreeSurfer (version 5.3) (*Fischl, 2012*), and cortical and subcortical gray matter was parcellated in native structural space according to the Lausanne atlas (*Cammoun et al., 2012*), which includes whole-brain sub-divisions of the Desikan-Killany anatomical atlas (*Desikan et al., 2006*) at multiple spatial scales. The acquired 233-region gray matter parcellation of each subject was dilated by 2 mm and then masked by the boundary of each subject's white matter segmentation (*Baum et al., 2018*). Once defined for each subject, the structural parcellation atlas was co-registered to the first b = 0 vol of each subject's diffusion image using boundary-based registration (*Greve and Fischl, 2009*). These parcels were then used as nodes for brain network construction. The left lateral occipital parcel was missing in 18 subjects and therefore was removed from analyses, yielding 232 brain regions that were present in all participants.

### Diffusion image pre-processing

FSL was used for diffusion data processing (*Jenkinson et al., 2012*; *Smith et al., 2004*). The two consecutive 32-direction acquisitions were merged into a single 64-direction time series. In-scanner head motion and the total network strength were used as covariates in this study. Specifically, in-scanner head motion was measured by the mean relative volume-to-volume displacement between the higher SNR b = 0 images (n = 7), which summarizes the total translation and rotation in 3-dimensional Euclidean space (*Baum et al., 2018*; *Roalf et al., 2016*). A mask in subject diffusion space was defined by registering a binary mask of a standard fractional anisotropy (FA) map (FMRIB58 FA) to each subject's dMRI reference image (mean b = 0) using FSL *FLIRT*. This mask was provided as input to FSL *eddy* in addition to the non-brain extracted dMRI image. Eddy currents and subject motion were estimated and corrected using the FSL *eddy* tool (version 5.0.5: [*Andersson and Sotiropoulos, 2016*]). This procedure uses a Gaussian Process to simultaneously model the effects of eddy currents and head motion on diffusion-weighted volumes, resampling the data only once. Diffusion gradient vectors were also rotated to adjust for subject motion estimated by eddy (*Leemans and Jones, 2009*). After the field map was estimated, distortion correction was then applied to dMRI images using FSL's *FUGUE* utility.

### Probabilistic tractography and network construction

We first fitted a ball-and-sticks diffusion model for each subject's dMRI data with FSL *bedpostx*, which uses Markov chain Monte Carlo sampling to build distributions on principal fiber orientation and diffusion parameters at each voxel (*Behrens et al., 2007*). Probabilistic tractography was run using FSL *probtrackx*, which repetitively samples voxel-wise fiber orientation distributions to model the spatial trajectory and strength of anatomical connectivity between specified seed and target regions (*Behrens et al., 2007*).

Each cortical and subcortical region defined along the gray-white boundary was selected as a seed region, and its connectivity strength to each of the other 231 regions was calculated using probabilistic tractography. At each seed voxel, 1000 samples were initiated. We used the default tracking parameters (a step-length of 0.5 mm, 2000 steps maximum, curvature threshold of 0.02). To increase the biological plausibility of white matter pathways reconstructed with probabilistic tractography, streamlines were terminated if they traveled through the pial surface, and discarded if they traversed cerebro-spinal fluid (CSF) in ventricles or re-entered the seed region (*Baum et al., 2018*). The connection probability from the seed voxel *i* to another voxel *j* was defined by the number of fibers passing through voxel *j* divided by the total number of fibers that were not rejected by exclusion criteria sampled from voxel *i*. For a seed cortical region, $1,000 \times n$ fibers were sampled (1000 fibers per voxel), where n is the number of voxels in this region. The number of fibers passing through a given region divided by $1,000 \times n$ is calculated as the connectivity probability from the seed region to this given region. Therefore, a 232*232 connection probability matrix was created for each subject. Notably, the probability from region *i* to region *j* is not necessarily equivalent to the one from region *j* to region *i* due to the dependence of tractography on the seeding location. Thus, we defined the unidirectional connectivity probability $P_{ij}$ between region *i* and region *j* by averaging these two probabilities (*Baum et al., 2018*; *Gong et al., 2009a*).

## Defining a priori network modules

Each of the 232 nodes in our network was assigned to a standard set of 7 functional systems originally defined by *Yeo et al. (2011)* in a whole-brain clustering analysis. To make this assignment, we calculated the purity index for the 7-system parcellation and brain regions from the Lausanne 232 parcellation atlas as in prior work (*Baum et al., 2017*). This measure quantifies the maximum overlap of cortical Lausanne labels and functional systems defined by *Yeo et al. (2011)*. Each cortical Lausanne label was assigned to a functional system by calculating the non-zero mode of all voxels in each brain region (*Figure 1—figure supplement 2*). Subcortical regions were assigned to an eighth, subcortical module.

## Control analysis

We investigated how a structural brain network composed of white matter fiber tracts constrains the brain in transitioning from a baseline state (i.e., 1 × 232 zero vector) to a fronto-parietal activation state, which was defined as regions in the fronto-parietal system that had activity magnitude equal to one while other regions had activity magnitude equal to 0. According to previous studies (*Betzel et al., 2016*; *Gu et al., 2017*; *Kim et al., 2018*; *Stiso et al., 2019*), we employed a simplified noise-free linear continuous-time and time-invariant network model:

$$\dot{x}(t) = \mathbf{A}x(t) + \mathbf{B}u(t) \tag{1}$$

Here, $x(t)$ is a 1 × N vector that represents the brain state at a given time, where N is the number of ROIs (N = 232). The initial sate $x(0)$ is a 1 × 232 zero vector, and the target state $x_T$ is a 1 × 232 vector of fronto-parietal activation. The matrix **A** encodes the connection probability weighted network, where **A** has been scaled by its largest eigenvalue and had the identity matrix subtracted to assure that it is stable (*Betzel et al., 2016*; *Gu et al., 2017*; *Karrer et al., 2020*; *Stiso et al., 2019*). The matrix **B** is a N × N input matrix that identifies the nodes in the control set. Here, **B** is an identity matrix because all 232 regions in the whole brain were control nodes. The input $u(t)$ denotes the control energy injected for each node at a given time.

This work aims to model the control process necessary to activate the fronto-parietal system, which is critical to executive function. We set the baseline state to zero, because we sought to model the contrast in activation between an executive task and the resting state. This comparison is motivated by a long history of task fMRI experiments that explicitly contrast executive tasks to the resting state, resulting in robust activation of the fronto-parietal cortex (*Cohen et al., 1997*; *Forsyth et al., 2014*; *Nagel et al., 2009*; *Ragland et al., 2002*; *Rowe et al., 2000*). We set the values of regions in the fronto-parietal system to one to represent the fact that these regions were activated.

We were interested in a control task where the system transitions from initial state $x(0)$ to target state $x_T$ with minimum-energy input, which is an optimal control problem. We first defined a cost function as the weighted sum of the energy cost of the transition and the integrated squared distance between the transition states and the target state.

$$\min_u \int_0^T (x_T - x(t))^T \mathbf{S}(x_T - x(t)) + \rho u(t)^T u(t) dt, \tag{2}$$
$$s.t. \quad \dot{x}(t) = \mathbf{A}x(t) + \mathbf{B}u(t), \ x(0) = x_0, \text{ and } x(T) = x_T,$$

where $x_T$ is the target state, $(x_T - x(t))^T(x_T - x(t))$ is the distance between the state at time $t$ and the target state $x_T$, T is a free parameter that defines the finite amount of time given to reach the target state, and $\rho$ is a free parameter that weights the energy constraint. Because the time of each step was defined as 0.001, there were 1,000 steps from initial to target state if we set T=1. **S** is 0-1 diagonal matrix of size N×N that selects only the nodes that we wish to control. Here, we only constrain the activity of the fronto-parietal system. Specially, $(x_T - x(t))^T \mathbf{S}(x_T - x(t))$ constrains the trajectories of all nodes in fronto-parietal system by preventing the system from traveling too far from the target state, and $u(t)^T u(t)$ constrains the amount of energy used to reach the target state.

To compute an optimal $u^*$ that induces a transition from the initial state $x(0)$ to the target state $x_T$, we define a Hamiltonian as:

$$H(p, x, u, t) = (x_T - x)^T \mathbf{S}(x_T - x) + \rho u^T u + p(\mathbf{A}x + \mathbf{B}u) \tag{3}$$

From the Pontryagin minimum principle (*Boltyanskii et al., 1960*), if $u^*$ is a solution to the minimization problem with corresponding trajectory $x^*$, then there exists $p^*$ such that:

$$\frac{\partial H}{\partial x} = -2\boldsymbol{S}(x_T - x^*) + \boldsymbol{A}^T p^* = -\dot{p}^*, \tag{4}$$

$$\frac{\partial H}{\partial u} = 2\rho u^* + \boldsymbol{B}^T p^* = 0. \tag{5}$$

From *Equation (5)* and *Equation (1)*, we derive that

$$u^* = -\frac{1}{2\rho}\boldsymbol{B}^T p^*, \tag{6}$$

$$\dot{x}^* = \boldsymbol{A}x^* - \frac{1}{2\rho}\boldsymbol{B}\boldsymbol{B}^T p^*. \tag{7}$$

Then, we rewrite *Equations (4) and (7)* as

$$\begin{bmatrix} \dot{x}^* \\ \dot{p}^* \end{bmatrix} = \begin{bmatrix} \boldsymbol{A} & -\frac{1}{2\rho}\boldsymbol{B}\boldsymbol{B}^T \\ -2\boldsymbol{S} & -\boldsymbol{A}^T \end{bmatrix} \begin{bmatrix} x^* \\ p^* \end{bmatrix} + \begin{bmatrix} 0 \\ 2\boldsymbol{S} \end{bmatrix} x^T, \tag{8}$$

We denote:

$$\tilde{\boldsymbol{A}} = \begin{bmatrix} \boldsymbol{A} & -\frac{1}{2\rho}\boldsymbol{B}\boldsymbol{B}^T \\ -2\boldsymbol{S} & -\boldsymbol{A}^T \end{bmatrix},$$

$$\tilde{x} = \begin{bmatrix} x^* \\ p^* \end{bmatrix},$$

$$\tilde{\boldsymbol{b}} = \begin{bmatrix} 0 \\ 2\boldsymbol{S} \end{bmatrix} x_T,$$

Then, *Equation (8)* can be reduced as:

$$\dot{\tilde{x}} = \tilde{\boldsymbol{A}}\tilde{x} + \tilde{\boldsymbol{b}},$$

Which can be solved as:

$$\tilde{x}(t) = e^{\tilde{\boldsymbol{A}}t}\tilde{x}(0) + \tilde{\boldsymbol{A}}^{-1}\left(e^{\tilde{\boldsymbol{A}}t} - \boldsymbol{I}\right)\tilde{\boldsymbol{b}}. \tag{9}$$

Then, by fixing t = T, we rewrote *Equation (9)* as

$$\tilde{x}(T) = e^{\tilde{\boldsymbol{A}}T}\tilde{x}(0) + \tilde{\boldsymbol{A}}^{-1}\left(e^{\tilde{\boldsymbol{A}}T} - \boldsymbol{I}\right)\tilde{\boldsymbol{b}}. \tag{10}$$

Let

$$c = \tilde{\boldsymbol{A}}^{-1}\left(e^{\tilde{\boldsymbol{A}}T} - \boldsymbol{I}\right)\tilde{\boldsymbol{b}},$$

$$e^{\tilde{\boldsymbol{A}}T} = \begin{bmatrix} \boldsymbol{E}_{11} & \boldsymbol{E}_{12} \\ \boldsymbol{E}_{21} & \boldsymbol{E}_{22} \end{bmatrix}.$$

We can then rewrite *Equation (10)* as:

$$\begin{bmatrix} x^*(T) \\ p^*(T) \end{bmatrix} = \begin{bmatrix} \boldsymbol{E}_{11} & \boldsymbol{E}_{12} \\ \boldsymbol{E}_{21} & \boldsymbol{E}_{22} \end{bmatrix} \begin{bmatrix} x^*(0) \\ p^*(0) \end{bmatrix} + \begin{bmatrix} c_1 \\ c_2 \end{bmatrix},$$

from which we can obtain

$$x^*(T) = \boldsymbol{E}_{11}x^*(0) + \boldsymbol{E}_{12}p^*(0) + c_1,$$

which can be rearranged to

$$p^*(0) = \boldsymbol{E}_{12}^{-1}[x^*(T) - \boldsymbol{E}_{11}x^*(0) - c_1].$$

Now that we have obtained *p\*(0)*, we can use it and *x(0)* to solve for $\tilde{x}$ via forward integration according to **Equation (9)**. To solve for *u\**, we take *p\** from our solution of $\tilde{x}$ and plug it into **Equation (6)**.

To quantify differences in trajectories, and the ease of controlling the system, we calculated a single measure of energy for every trajectory. Particularly, the energy of each control node *i* was defined as:

$$E_i = \int_{t=0}^{T} ||u_i^*(t)||^2.$$

## Comparison to null model network

In order to determine whether the topology of brain networks specifically facilitated transitions to the fronto-parietal activation target state, we compared the energetic cost to that of null model networks. Specifically, for each participant we constructed 100 null model networks where the degree and strength distribution was preserved (*Rubinov and Sporns, 2010*). We compared the control energy cost of the transition to the fronto-parietal activation target state estimated from the empirical networks to the average energy cost estimated in these null networks using a paired *t*-test.

## Statistical analyses of developmental and cognition effects

Prior studies demonstrated that the developmental changes of brain structure and function could be either linear (*Hagmann et al., 2010*; *Wierenga et al., 2016*) or non-linear (*Grayson and Fair, 2017*; *Mills et al., 2016*; *Vandekar et al., 2015*). Accordingly, for our developmental analyses we used generalized additive models (GAMs) in order to simultaneously model linear and nonlinear relationships with age using penalized splines (*Wood, 2004*). We evaluated associations between control energy and age at multiple resolutions, including the whole brain, cognitive systems, and network nodes. Similarly, we evaluated associations between control energy and executive performance while controlling for age. For all models, we included sex, handedness, total brain volume, total network strength, and in-scanner head motion during the diffusion scan as model covariates. Multiple comparisons were accounted for using the False Discovery Rate (*q* < 0.05). For developmental effect of control energy, the GAM model was:

Energy = spline(Age) + Sex + Handedness + Motion + TBV + Network Strength + intercept.
For the associations between control energy and executive performance, the GAM model was:
Energy = Executive Performance + spline(Age) + Sex + Handedness + Motion + TBV + Network Strength + intercept.

We used the *gam* command in the R package '*mgcv*' to implement the model. The spline term estimates a nonparametric smooth function for age-related differences in control energy, which can include linear or nonlinear effects depending on the structure of the data. Restricted maximum likelihood is used to penalize non-linearity in order to prevent overfitting (*Wood, 2004*).

Furthermore, for regions that displayed the associations between control energy and both age and cognition, we evaluated whether regional control energy might mediate the relationship between age and executive function. Specifically, we regressed out the effects of nuisance covariates (i.e., sex, handedness, total brain volume, total network strength, and in-scanner head motion) on the independent (X, age), dependent (Y, executive efficiency) and mediating (M, control energy) variables using a linear model. The resultant normalized residuals were used in our mediation analysis. We then evaluated the significance of the indirect effect using bootstrapped confidence intervals within the R package *lavaan*. Then, we examined: 1) path **c**: the total effect of age on executive performance; 2) path **a**: the relationship between age and the control energy; 3) path **b**: the relationship between control energy and executive performance; and 4) path **c'**: the age effect of executive function controlling for the mediator/control energy. The mediation/indirect effect **a*b** is the effect size of the relationship between age and executive performance that was reduced after controlling for

the mediator/control energy. For each path, we calculated the beta coefficient, which reflected the changes of the outcome for every one-unit change in the predictor. A bootstrap analysis (i.e., resampled 10,000 times) was implemented to estimate the confidence intervals for the indirect effect.

## Prediction of brain maturity from the pattern of control energy

As a complement to the mass-univariate analyses described above, we also sought to predict individual brain maturity using the multivariate pattern of control energy (*Dosenbach et al., 2010*; *Erus et al., 2015*; *Franke et al., 2012*). We used ridge regression with nested two-fold cross validation (2F-CV).

### Ridge regression

A linear regression model was adopted to predict brain maturity using the pattern of whole-brain control energy. The linear model can be formalized as follows:

$$y_i = \sum_{j=1}^{p} \beta_j x_{i,j} + \beta_0,$$

where $y_i$ is the age of the $i^{th}$ individual, $p$ is the number of features, $x_{i,j}$ is the value of the $j^{th}$ feature of the $i^{th}$ subject, and $\beta_j$ is the regression coefficient.

To avoid over-fitting and to improve the prediction accuracy, we applied ridge regression (*Cui and Gong, 2018*; *Hoerl and Kennard, 1970*; *Siegel et al., 2016*), which used an L2 penalty during model fitting. The objective function is:

$$\min_{\beta} \sum_{i=1}^{N} (f(x_i) - y_i)^2 + \lambda \sum_{j=1}^{p} ||\beta_j||^2.$$

This technique shrinks the regression coefficients, resulting in better generalizability for predicting unseen samples. In this algorithm, a regularization parameter $\lambda$ is used to control the trade-off between the prediction error of the training data and L2-norm regularization, i.e., a trade-off of penalties between the training error and model complexity. A large $\lambda$ corresponds to a greater penalty on model complexity, and a small $\lambda$ represents a greater penalty on training error. Compared with the traditional ordinary least squares regression, ridge regression is less impacted by multicollinearity and can avoid over-fitting (*Cui and Gong, 2018*).

### Prediction framework

See *Figur 3—figure supplement 1* for the schematic overview of the prediction framework. Specifically, we applied a nested 2-fold cross validation (2F-CV), with outer 2F-CV estimating the generalizability of the model and the inner 2F-CV determining the optimal parameter $\lambda$ for the ridge regression model.

### Outer 2F-CV

In the outer 2F-CV, all subjects were divided into 2 subsets. Specifically, we sorted the subjects according to their age and then assigned the individuals with an odd rank to subset 1 and the individuals with an even rank to subset 2 (*Cui and Gong, 2018*; *Cui et al., 2018*). We first used subset 1 as a training set, and we used subset 2 as a testing set. Each feature was linearly scaled between zero and one across the training dataset, and the scaling parameters were also applied to scale the testing dataset (*Cui and Gong, 2018*; *Cui et al., 2018*). We applied an inner 2-fold cross validation (2F-CV) within training set to select the optimal $\lambda$ parameter. Based on the optimal $\lambda$, we trained a model using all subjects in the training set, and then used that model to predict the age of all subjects in the testing set. Analogously, we used subset 2 as a training set and subset 1 as a testing set, and repeated the above procedure. Across the testing subjects for each fold, the correlation and mean absolute error (MAE) between the predicted and actual age was used to quantify the prediction accuracy. Here, we used the scikit-learn library to implement ridge regression (http://scikit-learn.org) (*Pedregosa et al., 2011*).

### Inner 2F-CV

Within each loop of the outer 2F-CV, we applied inner 2F-CVs to determine the optimal $\lambda$. Specially, the training set for each loop of the outer 2F-CV was further partitioned into 2 subsets according to their rank of the age, as like the outer loop (i.e., subjects with odd rank in subset 1 and subjects with even rank in subset 2). One subset was selected to train the model under a given $\lambda$ in the range $[2^{-10}, 2^{-9}, ..., 2^4, 2^5]$ (i.e., 16 values in total) (*Cui and Gong, 2018*), and the remaining subset was used to test the model. This procedure was repeated 2 times such that each subset was used once as the testing dataset, resulting in 2 inner 2F-CV loops in total. For each inner 2F-CV loop, the correlation $r$ between the actual and predicted age and the mean absolute error (MAE) were calculated for each $\lambda$, and averaged over each fold. The sum of the mean correlation $r$ and reciprocal of the mean MAE was defined as the inner prediction accuracy, and the $\lambda$ with the highest inner prediction accuracy was chosen as the optimal $\lambda$ (*Cui and Gong, 2018*; *Cui et al., 2018*). Of note, the mean correlation $r$ and the reciprocal of the mean MAE cannot be summed directly, because the scales of the raw values of these two measures are quite different. Therefore, we normalized the mean correlation $r$ and the reciprocal of the mean MAE across all values and then summed the resultant normalized values.

### Evaluation of generalizability

The correlation and mean absolute error (MAE) between the predicted 'brain age' and chronological age was used to quantify the degree to which the model captured the development trajectory of the brain. Particularly, when we calculated the correlation, we controlled sex, handedness, total brain volume, total network strength, and in-scanner head motion.

### Interpreting the model

In a linear prediction model such as ridge regression, one weight/regression coefficient was assigned for each feature/brain region. We trained a prediction model using all the samples and acquired a weight vector $\boldsymbol{w}$. According to *Haufe et al. (2014)*; *Waskom and Wagner (2017)*, we left multiplied the model weight vector $\boldsymbol{w}$ by the data covariance matrix $\sum_X$, which was formulized as $\boldsymbol{a} = \sum_X \cdot \boldsymbol{w}$. The transformed weight vector $\boldsymbol{a}$ was a distributed pattern quantifying the contribution of each brain region in the multivariate ridge prediction. The absolute value of the transformed weight represents the importance of the corresponding feature in a prediction (*Haufe et al., 2014*; *Mourão-Miranda et al., 2005*).

### Randomly split 2F-CV

In the above prediction analysis, we split subjects into two halves according to their age rank. For completeness, we also split the subjects randomly into two halves for both outer 2F-CV and inner 2F-CV, and calculated the mean partial correlation $r$ and MAE across two folds. Because the split is random, we repeated this procedure 100 times and averaged the partial correlation and MAE across the 100 times to acquire the final prediction accuracy.

## Specificity and sensitivity analysis

We conducted several additional supplementary analyses to assess the sensitivity and specificity of our results. First, in order to evaluate the robustness of our results to variation in target states, we additionally generated 100 new initial states and 100 new target states with noise added. In this distribution of initial states, the activation value of regions in the fronto-parietal system is Gaussian with a mean value of 0 and a standard deviation of 0.1, while in the distribution of target states, the activation value of regions in the fronto-parietal system is Gaussian with a mean value of 1 and a standard deviation of 0.1. Second, to ensure the observed associations with age were driven by the topological structure of real brain networks, we tested whether age effects existed using null networks that preserved the degree and strength distribution. We created 100 null networks and calculated the one-tailed $P$ value for effect size of whole-brain and fronto-parietal system, which was the portion of null networks that showed a lower negative effect size value than the actual value for the real network. Third, we assessed whether the structural network also contributed to other cognitive functions as well by comparing the control energy cost required to reach a motor activation state for real networks and null networks. We further evaluated the age effects of control energy cost needed to activate the motor system.

Fourth, the present work explored a specific transition of the brain from a baseline state to a state of fronto-parietal activation by enacting multi-point control. In contrast, modal controllability quantifies the difficulties of transitioning to all possible states via single-node control (*Gu et al., 2015*). Modal controllability identifies brain areas that can push the brain into difficult-to-reach states; our prior work has shown that modal controllability increases with age in youth (*Tang et al., 2017*). Accordingly, it is important to establish whether our present results were driven by developmental changes in modal controllability. As in *Tang et al. (2017)*, before calculating controllability, we scaled the matrix by $1+\xi_0$, where $\xi_0$ is the largest eigenvalue value of the matrix. Next, we conducted sensitivity analyses where we controlled for modal controllability by including it as a covariate in the regression equation at each resolution of analysis (e.g., whole brain, functional system, network nodes). Specifically, we controlled for nodal modal controllability in nodal analysis of control energy, controlled for the average modal controllability of each system for system-level analysis, and controlled for the whole-brain average modal controllability for whole-brain analysis.

Fifth, one might expect that the modular organization of the brain's structural network could potentially change the control energy cost of brain state transitions (*Avena-Koenigsberger et al., 2017*). Prior work has reported age-related increases in brain network modularity during youth (*Baum et al., 2017*; *Hagmann et al., 2010*; *Huang et al., 2015*). Here, we evaluated if observed developmental associations with control energy might be driven by changes in network modularity. We calculated network modularity quality (*Q*) using the community structure defined by the functional atlas (*Yeo et al., 2011*) as in *Baum et al. (2017)*. For comparability with analyses of control energy, we scaled the matrix by the maximum eigenvalue before calculating *Q*. However, for this specific analysis, we did not subtract the identity matrix because it would lead to (uninterpretable) negative values of *Q*. We controlled for *Q* by including it as a model covariate in sensitivity analyses, which were conducted at all resolutions (whole brain, functional systems, and network nodes). Further, we evaluated the possibility that the segregation of the fronto-parietal system during youth (*Baum et al., 2017*) could explain the age effect of control energy. We calculated the average participation coefficient of the fronto-parietal system, and evaluated if developmental associations with control energy in the fronto-parietal system remained while controlling for the average participation coefficient in this system alongside with other covariates.

Finally, we evaluated an alternative, biologically recorded target state that was defined using the activation pattern from a working memory task that reliably recruits the fronto-parietal network and executive system. Specifically, the target state was defined as the average participants' 2-back > 0-back contrast from a fractal *n*-back working memory task (*Ragland et al., 2002*); task design and image processing was as previously detailed (*Satterthwaite et al., 2013*). For each participant, we calculated the control energy cost to transition from the baseline (zero) state to this target activation state defined by the average pattern of activation recruited by the working memory task. As in the main analyses, we compared the control energy cost from real brain networks and that from null networks. Furthermore, we calculated the developmental association between control energy and age at multiple scales, including the whole brain, each cognitive system, and for each network node (with covariates as prior).

## Acknowledgements

Thanks to Chad Jackson, in memoriam. This study was supported by grants from National Institute of Mental Health: R21MH106799 (DSB and TDS) and R01MH113550 (TDS and DSB). Additional support was provided by R01MH107703 (TDS), RF1MH116920 (DJO, TDS and DSB), R01MH112847 (RTS and TDS), R01MH107235 (RCG), and R01EB022573 (CD), K01MH102609 (DRR), R01NS085211 (RTS), and the Penn-CHOP Lifespan Brain Institute. The PNC was supported by MH089983 and MH089924. Additionally, DSB acknowledges support from the John D and Catherine T MacArthur Foundation, the Alfred P Sloan Foundation, the ISI Foundation, the Paul Allen Foundation, the Army Research Laboratory (W911NF-10-2-0022), the Army Research Office (Bassett-W911NF-14-1-0679, Grafton-W911NF-16-1-0474, DCISTW911NF-17-2-0181), the Office of Naval Research, National Institute of Health (2-R01-DC-009209–11, 1R01HD086888-01, R01 – MH112847, R01-MH107235), National Institute of Neurological Disorders and Stroke (R01 NS099348), and National Science Foundation (BCS-1441502, BCS-1430087, NSF PHY-1554488 and BCS- 1631550). The content is solely

the responsibility of the authors and does not necessarily represent the official views of any of the funding agencies.

## Additional information

### Competing interests

Russell T Shinohara: has received legal consulting and advisory board income from Genentech/Roche. The other authors declare that no competing interests exist.

### Funding

| Funder | Grant reference number | Author |
|---|---|---|
| National Institute of Mental Health | R21MH106799 | Danielle S Bassett<br>Theodore D Satterthwaite |
| National Institute of Mental Health | R01MH113550 | Theodore D Satterthwaite<br>Danielle S Bassett |
| National Institute of Mental Health | R01MH107703 | Theodore D Satterthwaite |
| National Institute of Mental Health | RF1MH116920 | Desmond J Oathes |
| National Institutes of Health | R01MH112847 | Russell T Shinohara<br>Theodore D Satterthwaite |
| National Institutes of Health | R01MH107235 | Ruben C Gur |
| National Institutes of Health | R01EB022573 | Christos Davatzikos |
| National Institutes of Health | K01MH102609 | David R Roalf |
| National Institutes of Health | R01NS085211 | Russell T Shinohara |
| John D and Catherine T MacArthur Foundation | | Danielle S Bassett |
| Alfred P. Sloan Foundation | | Danielle S Bassett |
| ISI Foundation | | Danielle S Bassett |
| Paul G. Allen Family Foundation | | Danielle S Bassett |
| Army Research Laboratory | W911NF-10-2-0022 | Danielle S Bassett |
| Army Research Office | Bassett-W911NF-14-1-0679 | Danielle S Bassett |
| Army Research Office | Grafton-W911NF-16-1-0474 | Danielle S Bassett |
| Army Research Office | DCISTW911NF-17-2-0181 | Danielle S Bassett |
| Office of Naval Research | | Danielle S Bassett |
| National Institutes of Health | 2-R01-DC-009209–11 | Danielle S Bassett |
| National Institutes of Health | 1R01HD086888-01 | Danielle S Bassett |
| National Institutes of Health | R01-MH112847 | Danielle S Bassett |
| National Institutes of Health | R01-MH107235 | Danielle S Bassett |
| National Institute of Neurological Disorders and Stroke | R01 NS099348 | Danielle S Bassett |
| National Science Foundation | BCS-1441502 | Danielle S Bassett |
| National Science Foundation | BCS-1430087 | Danielle S Bassett |
| National Science Foundation | NSF PHY-1554488 | Danielle S Bassett |
| National Science Foundation | BCS-1631550 | Danielle S Bassett |

The funders had no role in study design, data collection and interpretation, or the decision to submit the work for publication.

### Author contributions

Zaixu Cui, Conceptualization, Software, Formal analysis, Investigation, Visualization, Methodology, Project administration; Jennifer Stiso, Validation, Methodology; Graham L Baum, Data curation, Software; Jason Z Kim, Richard F Betzel, Shi Gu, Zhixin Lu, Xiaosong He, Fabio Pasqualetti, Methodology; David R Roalf, Rastko Ciric, Tyler M Moore, Data curation; Cedric H Xia, Visualization; Desmond J Oathes, Russell T Shinohara, Kosha Ruparel, Christos Davatzikos, Raquel E Gur, Ruben C Gur, Resources; Danielle S Bassett, Conceptualization, Resources, Software, Supervision, Funding acquisition, Methodology, Project administration; Theodore D Satterthwaite, Conceptualization, Resources, Data curation, Software, Supervision, Funding acquisition, Investigation, Methodology, Project administration

### Author ORCIDs

Zaixu Cui (iD) https://orcid.org/0000-0003-4385-8106
Jennifer Stiso (iD) http://orcid.org/0000-0002-3295-586X
Desmond J Oathes (iD) http://orcid.org/0000-0001-7346-2669
Ruben C Gur (iD) https://orcid.org/0000-0002-9657-1996
Danielle S Bassett (iD) https://orcid.org/0000-0002-6183-4493
Theodore D Satterthwaite (iD) https://orcid.org/0000-0001-7072-9399

### Ethics

Human subjects: All subjects or their parent/guardian provided informed consent, and minors provided assent. The Institutional Review Boards of both Penn and CHOP approved study procedures (IRB-approved protocol number 810336).

### Decision letter and Author response

Decision letter https://doi.org/10.7554/eLife.53060.sa1
Author response https://doi.org/10.7554/eLife.53060.sa2

## Additional files

### Supplementary files

• Transparent reporting form

### Data availability

The PNC data is publicly available in the Database of Genotypes and Phenotypes: accession number: phs000607.v3.p2; https://www.ncbi.nlm.nih.gov/projects/gap/cgi-bin/study.cgi?study_id=phs000607.v3.p2. All analysis code is available here: https://github.com/ZaixuCui/pncControlEnergy (copy archived at https://github.com/elifesciences-publications/pncControlEnergy), with detailed explanation in https://github.com/ZaixuCui/pncControlEnergy/wiki.

The following previously published dataset was used:

| Author(s) | Year | Dataset title | Dataset URL | Database and Identifier |
|---|---|---|---|---|
| Satterthwaite TD, Elliott MA, Ruparel K, Loughead J, Prabhakaran K, Calkins ME, Hopson R, Jackson C, Keefe J, Riley M, Mentch FD, Sleiman P, Verma R, Davatzikos C, Ha- | 2014 | Neuroimaging of the Philadelphia neurodevelopmental cohort | https://www.ncbi.nlm.nih.gov/projects/gap/cgi-bin/study.cgi?study_id=phs000607.v3.p2 | NCBI, phs000607.v3.p2 |

konarson H, Gur
RC, Gur RE

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
