## [Decision Letter]

**Acceptance summary:**

In this study, Cui and colleagues utilized linear control theory to compute the amount of "energy" necessary for a brain to transition from a baseline state to a frontoparietal control state. They showed that minimum control energy necessary for the transition is lower for older participants and is maximal in the frontoparietal regions. The patterns of control energy can be used to predict participants' age. Furthermore, control energy of cingulate cortex is negatively correlated with executive function performance even after controlling for age. This works highlights a potential mechanism by which executive function develops. The work is technically excellent and the findings are likely to be of broad interest to the community. Moreover, the authors demonstrate the effect from multiple angles. We are also very impressed by the authors' response to the reviewers' comments.

**Decision letter after peer review:**

Thank you for submitting your article "Optimization of energy state transition trajectory supports the development of executive function during youth" for consideration by *eLife*. Your article has been reviewed by three peer reviewers, one of whom is a member of our Board of Reviewing Editors, and the evaluation has been overseen by Timothy Behrens as the Senior Editor. The reviewers have opted to remain anonymous.

The reviewers have discussed the reviews with one another and the Reviewing Editor has drafted this decision to help you prepare a revised submission.

Summary:

In this study, Cui and colleagues utilized linear control theory to compute the amount of "energy" necessary for a brain to transition from a baseline state (zero activity everywhere) to a frontoparietal control state (activity of ones in frontoparietal regions and zeros everywhere else). They showed that minimum control energy necessary for the transition is lower for older participants and is maximal in the frontoparietal regions. The patterns of control energy can be used to predict participants' age. Furthermore, control energy of cingulate cortex is negatively correlated with executive function performance even after controlling for age. This works highlights a potential mechanism by which executive function develops. The manuscript follows logically from the authors' previous work and convincingly demonstrates that network-mediated control of executive areas is correlated with age. The work is technically excellent and the findings are likely to be of broad interest to the community. Moreover, the authors demonstrate the effect from multiple angles.

Essential revisions:

1) For a life sciences journal, the manuscript is quite technical and uses a lot of jargon such as modal controllability that is not well defined and might not be understandable to a life science readership. It would be useful to better explain the biological basis of terms such as the control trajectory distance.

2) We suggest toning down language throughout the paper. Associations of r=0.17 can hardly be called strong, particularly given that this is cross-sectional sample. The extended discussion of electrical stimulation and neurofeedback is peripheral to the current results, given that these modalities were not investigated. I suggest keeping the discussion specific to the current results and providing some basic discussion of biological or network mechanisms that could potentially underpin the relation between control energy and age.

3) While network control theory is a novel way to capture the relation between brain network development, it would be useful to understand the basic network properties that change over age and therefore underpin the relation between age and control energy. As they currently stand, the control theory results are quite abstract and do not appear to provide insight into specific network-related or biological mechanisms that enable lower-cost transitions.

a) The computation of control energy is fully characterized by the network connectivity matrix. Therefore, any relation between age and control energy should be able to be traced back to relations between age and other simpler network properties of the connectivity matrix. It would be informative to evaluate whether the modular structure or connectivity strength relates to age, which could provide a simpler characterization of the control energy result that is closer to the underlying biology.

b) One possibility is that the FP network may show greater modularization with age, together with an overall increase in network integration. Indeed, the authors have previously suggested increased segregation of the FPN in this dataset (Baum et al. 2017). The fact that control energy is greater in the randomized null might be consistent with this notion. This kind of insight would be useful to link the control energy results and underlying network-level mechanisms.

c) A related possibility is that the results might simply be re-capitulating the fact that younger participants have weaker long range connections. First, it seems obvious that the best way to "activate" the frontoparietal regions would be to inject energy into the target (frontoparietal) regions, so it's not surprising that frontoparietal regions required the most control energy (Figure 1C). For older participants, this energy might be lower because frontoparietal regions are now more strongly connected to other regions, so the overall control energy (and frontoparietal control energy) can be reduced by "distributing" the control energy budget to other regions.

4) The authors should explicitly mention the definition of the final state and initial states in the Results section. While this is explained in the Materials and methods section, this should really be in the Results section because this is quite important.

5) The authors justified their choice of initial and final state as follows "We set the baseline state to zero, because we sought to model the contrast in activation between an executive task and the resting state. This comparison is motivated by a long history of task fMRI experiments that explicitly contrast executive tasks to the resting state, resulting in robust activation of the fronto-parietal cortex (Cohen et al., 1997; Forsyth et al., 2014; Nagel et al., 2009; Ragland et al., 2002; Rowe et al., 2000). We set the values of regions in the fronto-parietal system to 1 to represent the fact that these regions were activated." We agree with the rationale, but based on the rationale, it seems that a better initial state should be the fMRI activity pattern averaged across all time points during a resting-fMRI scan and the final state should be the fMRI activity pattern averaged across all time points during an executive function task.

6) Equation 2: "S is 0-1 diagonal matrix of size N x N that selects only the nodes that we wish to control. Here, we only constrain the activity of the fronto-parietal system." – Can the authors clarify this statement? Is S the identity matrix? We thought that all nodes are targeted since the target state comprises 1 for the frontoparietal nodes and 0 for other nodes. But the authors now seem to imply they only constrain the activity of the frontoparietal nodes.

7) "When model weights were examined at the level of individual network nodes, the regions that most contributed to the prediction of brain maturity aligned with univariate analyses, and included the dorsolateral and ventrolateral prefrontal cortex, the cingulate cortex, superior parietal cortex, and lateral temporal cortex" – It is well-known that model weights should not be interpreted without filtering (Haufe et al., 2014). Given the authors are using ridge regression, Haufe's approach is perfect for this situation.

8) The authors need to provide more details about how they controlled for linear and nonlinear effects of age in their analysis (e.g., correlation between control energy and executive performance). The authors mentioned generalized additive models and penalized splines, but the details were nowhere close to being sufficient.

9) The authors need to provide more details about the mediation analysis (e.g., assumptions). How do we interpret the betas in Figure 4C and D? Total effect is 0.67, while mediation effect is 0.03 and direct effect is 0.64. Doesn't this mean that the analysis is suggesting that the indirect effect is quite small relative to the direct effect? If so, this should be discussed.

10) The age prediction analysis requires motivation and better integration with the rest of the paper. It is likely that simpler features such as the tractography connectivity strengths or basic network properties would also yield good predictive models of age. Therefore, the reason for using a more complex feature space is unclear, unless it can be demonstrated that control energy can outperform the accuracy of more basic features, or reveals a specific mechanism that characterizes network development.

11) The relation identified between control energy and age appears to be linear based on Figure 2 and thus the reasons for fitting penalized splines to characterize nonlinear associations could be better described. Are the lines shown in Figure 2 representative of the splines? The claims regarding the specificity of the FP network require clarification. The null model corresponding to the rewired graph does not appear to have been evaluated for the other canonical networks. To establish specificity of the FP, the null model should also be considered for the other networks. Based on this global null model alone, we would suggest that it is hard to claim that the brain is explicitly wired to optimize transition cost to the FP activation state. Wiring organization is also likely to contribute to other cognitive functions as well.

12) In addition, it would be interesting to know whether the node-level measures are related to degree and/or strength. The authors use the two statistics as covariates in the predictive model, but do not show whether they are related to control energy directly.

---

## [Author Response]

Essential revisions:1) For a life sciences journal, the manuscript is quite technical and uses a lot of jargon such as modal controllability that is not well defined and might not be understandable to a life science readership. It would be useful to better explain the biological basis of terms such as the control trajectory distance.

We appreciate this comment, and agree. In the revised manuscript, we now explicitly define these terms, including control trajectory, state, trajectory distance, and modal controllability (Results section):

“Specifically, we defined the trajectory of a neural system to be the temporal path that the system traverses through diverse states, where the item state was defined as the vector of neurophysiological activity across brain regions at a single time point.”

“We calculated the trajectory distance at each time point, which was defined as the Euclidean distance between the current brain state and the target brain state. A small distance suggests that the current vector of brain activity is similar to the target vector of brain activity.”

“Fifth, we evaluated whether our developmental results could be explained by modal controllability. Modal controllability reflects the extent to which all dynamic modes of a system will change in response to small changes at a single node (Gu et al., 2015). If an individual has high modal controllability, it suggests that the underlying brain structural brain network was optimized to support efficient state transitions to diverse states. In line with this intuition, modal controllability increases with development in youth as flexible switching between patterns of brain activity becomes more common (Tang et al., 2017).”

2) We suggest toning down language throughout the paper. Associations of r=0.17 can hardly be called strong, particularly given that this is cross-sectional sample. The extended discussion of electrical stimulation and neurofeedback is peripheral to the current results, given that these modalities were not investigated. I suggest keeping the discussion specific to the current results and providing some basic discussion of biological or network mechanisms that could potentially underpin the relation between control energy and age.

We agree, and are happy to incorporate this feedback. We have moderated the language throughout the paper, removing any references to “strong associations” or “high accuracy.”

Furthermore, we have removed the discussion about electrical stimulation and neurofeedback. Instead, as suggested, we have expanded our discussion regarding the network mechanisms underlying the development of control energy (Discussion section):

“Our results suggest that neither the overall modularity of the structural network nor the segregation of the fronto-parietal system, which both mature during youth (Baum et al., 2017), could explain the association between age and control energy. […] In Srivastava et al., (2020), we show that the impulse response of the system is formally related to the network communicability, suggesting that paths of all lengths – not just direct connections – contribute to the control energy.”

3) While network control theory is a novel way to capture the relation between brain network development, it would be useful to understand the basic network properties that change over age and therefore underpin the relation between age and control energy. As they currently stand, the control theory results are quite abstract and do not appear to provide insight into specific network-related or biological mechanisms that enable lower-cost transitions.

This is a useful comment, and we are happy to address each specific point below.

a) The computation of control energy is fully characterized by the network connectivity matrix. Therefore, any relation between age and control energy should be able to be traced back to relations between age and other simpler network properties of the connectivity matrix. It would be informative to evaluate whether the modular structure or connectivity strength relates to age, which could provide a simpler characterization of the control energy result that is closer to the underlying biology.

This is a good point. Before calculating the control energy, to ensure stability, the matrix was been scaled by its largest eigenvalue and the identity matrix was subtracted as in prior work (Betzel et al., 2016; Gu et al., 2017; Karrer et al., 2020; Stiso et al., 2019). After scaling, the total connectivity strength weakly increased with age (Z = 2.22, P = 0.03, Partial r = 0.07, CI = [0.01, 0.14]). In all subsequent analyses we controlled for total network strength of this scaled network, including both mass-univariate analyses at multiple levels and multivariate pattern analysis. Accordingly, the strength of the network that was used to calculate the control energy could not explain the developmental decline of control energy observed.

Beyond network strength, our previous work has demonstrated that network modularity is related to age during youth (Baum et al., 2017). Accordingly, we calculated the network modularity quality (Q) using the community structure defined by the functional atlas (Yeo et al., 2011). As prior, we found that Q significantly increased with development (Z = 5.09, P = 3.62 × 10^-7^, Partial r = 0.17, CI = [0.10, 0.23]) while controlling for sex, handedness, motion, total brain volume, and total network strength. To determine whether network modularity Q explained our observed results, we calculated the developmental effects of control energy while also controlling for modularity. Results were generally consistent with our main results (Figure 2—figure supplement 3B). For example, average control energy of the whole brain and of the fronto-parietal system still significantly declined with age (whole-brain: Z = -3.95, P = 7.73 × 10^-5^, Partial r = -0.13, CI = [-0.19, -0.07]; fronto-parietal: Z = -4.31, P_FDR_ = 6.46 × 10^-5^, Partial r = -0.14, CI = [-0.20, -0.08]). These analyses suggest that network modularity does not explain the developmental decline of control energy in our results. We now include these analyses in the revised Materials and methods section and Results section.

Materials and methods section:

“Fifth, one might expect that the modular organization of the brain’s structural network could potentially change the control energy cost of brain state transitions (Avena-Koenigsberger et al., 2017). […] We controlled for Q by including it as a model covariate in sensitivity analyses, which were conducted at all resolutions (whole brain, functional systems, and network nodes).”

Results section:

“Sixth, because the modularity of brain networks evolves with age, one could ask whether that evolution impacts the observed assocations with control energy (Baum et al., 2017; Hagmann et al., 2010; Huang et al., 2015). However, we found that results remained consistent after controlling for network modularity in all analyses (Figure 2—figure supplement 3B). For example, average control energy of the whole brain and of the fronto-parietal system both significantly declined with age after controlling for network modularity (whole-brain: Z = -3.95, P = 7.73 × 10^-5^, Partial r = -0.13, CI = [-0.19, -0.07]; fronto-parietal: Z = -4.31, P_FDR_ = 6.46 × 10^-5^, Partial r = -0.14, CI = [-0.20, -0.08]).”

These results are also highlighted in revised Figure 2—figure supplement 3B.

b) One possibility is that the FP network may show greater modularization with age, together with an overall increase in network integration. Indeed, the authors have previously suggested increased segregation of the FPN in this dataset (Baum et al. *2017). The fact that control energy is greater in the randomized null might be consistent with this notion. This kind of insight would be useful to link the control energy results and underlying network-level mechanisms.*

This is a great suggestion. Above, we examined whether overall network modularity could explain our control energy results. To more specifically test the potential impact of fronto-parietal system segregation, we calculated the participation coefficient of each brain region in a manner consistent with our previous work (Baum et al., 2017), and averaged the participation coefficient of brain regions within the fronto-parietal system. Then, we evaluated the developmental changes of average control energy in the fronto-parietal system while controlling for average participation coefficient in the fronto-parietal system and other covariates. Results reveal that average control energy in the fronto-parietal system still significantly decreased with age (Z = -4.64, P = 3.51 × 10^-6^, Partial r = -0.15, CI = [-0.21, -0.09]), suggesting that increased segregation of the fronto-parietal system did not account for age associations with control energy. We have updated the Materials and methods section and Results section in the revised manuscript to reflect these new analyses:

Materials and methods section:

“Further, we evaluated the possibility that the segregation of the fronto-parietal system during youth (Baum et al., 2017) could explain the age effect of control energy. We calculated the average participation coefficient of the fronto-parietal system, and evaluated if developmental associations with control energy in the fronto-parietal system remained while controlling for the average participation coefficient in this system alongside with other covariates.”

Results section:

“We further assessed whether the increasing segregation of fronto-parietal system during youth (Baum et al., 2017) could explain the age effect of control energy. Results remained consistent after controlling for the average participation coefficient within the fronto-parietal system when examining age-related differences in control energy (Z = -4.64, P = 3.51 × 10^-6^, Partial r = -0.15, CI = [-0.21, -0.09]).”

c) A related possibility is that the results might simply be re-capitulating the fact that younger participants have weaker long range connections. First, it seems obvious that the best way to "activate" the frontoparietal regions would be to inject energy into the target (frontoparietal) regions, so it's not surprising that frontoparietal regions required the most control energy (Figure 1C). For older participants, this energy might be lower because frontoparietal regions are now more strongly connected to other regions, so the overall control energy (and frontoparietal control energy) can be reduced by "distributing" the control energy budget to other regions.

We appreciate this comment, and are happy to address this point. In order to examine connections both within the fronto-parietal system and between the fronto-parietal system and other systems, we calculated the sum of all the connections within the fronto-parietal system, as well as the sum of all the connections between the fronto-parietal system and other systems. Controlling for sex, handedness, motion, total brain volume and total network strength, we found that the within fronto-parietal system strength significantly increased (Z = 4.17, P = 3.05 × 10^-5^, Partial r = 0.14, CI = [0.07, 0.20]) during youth. Moreover, the total network strength between the fronto-parietal system and other systems also changed with age (Z = 3.27, P = 0.001, Partial r = 0.08, CI = [0.01, 0.14]).

Next, we assessed whether these effects could explain the observed developmental association with control energy. While controlling for within fronto-parietal system connectivity strength, the control energy in the fronto-parietal system still significantly declined with development (Z = -3.53, P = 0.0004, Partial r = -0.12, CI = [-0.18, -0.06]). Similarly, while controlling for the connectivity strength between fronto-parietal system and other systems, the control energy in the fronto-parietal system still significantly declined with development (Z = -4.88, P = 1.06 × 10^-6^, Partial r = -0.16, CI = [-0.22, -0.10]). These results suggest that connectivity either within the fronto-parietal system or between the fronto-parietal system and other systems does not explain age-related differences in control energy. We have added these results to the revised manuscript (Results section).

Results section:

“Seventh, we assessed whether connectivity within the fronto-parietal system or between the fronto-parietal and other systems could explain observed associations between age and control energy. […] Similarly, while controlling for the connectivity strength between the fronto-parietal system and other systems, the control energy in the fronto-parietal system still significantly declined with development (Z = -4.88, P = 1.06 × 10^-6^, Partial r = -0.16, CI = [-0.22, -0.10]).”

4) The authors should explicitly mention the definition of the final state and initial states in the Results section. While this is explained in the Materials and methods section, this should really be in the Results section because this is quite important.

We thank the reviewer for raising this point. We have added the definition of the initial and final states to the revised Results section.

Results section:

“Capitalizing on recent advances in network control theory, we modeled how structural networks facilitate state transitions from an initial baseline state to the target state. In the initial state, all regions had an activity magnitude of 0. In the target state, regions in the fronto-parietal system had activity magnitude of 1, with all other regions having an activity magnitude of 0.”

5) The authors justified their choice of initial and final state as follows "We set the baseline state to zero, because we sought to model the contrast in activation between an executive task and the resting state. This comparison is motivated by a long history of task fMRI experiments that explicitly contrast executive tasks to the resting state, resulting in robust activation of the fronto-parietal cortex (Cohen et al., 1997; Forsyth et al., 2014; Nagel et al., 2009; Ragland et al., 2002; Rowe et al., 2000). We set the values of regions in the fronto-parietal system to 1 to represent the fact that these regions were activated." We agree with the rationale, but based on the rationale, it seems that a better initial state should be the fMRI activity pattern averaged across all time points during a resting-fMRI scan and the final state should be the fMRI activity pattern averaged across all time points during an executive function task.

The reviewer raises a good point that the empirical BOLD signal could be used as an initial state. We fully acknowledge that these initial and target states are theoretical, and not obtained directly from imaging data. However, the choice of a simplified state allows us to isolate the energy required to activate the frontoparietal system without confounding effects of activity spread from other systems. Nonetheless, it is important that these theoretical states have biological plausibility. Therefore, as suggested, we first averaged the resting-state fMRI time series across all time points for several subjects and found the average value was close to 0 (e.g., around 7.63×10^-8^) for all brain regions. This indicated to us that we could retain theoretical simplicity without sacrificing too much biological plausibility. As such, we retained the baseline state as zero in all brain regions. However, in an effort to evaluate a more biologically plausible target state, we also evaluated a target state specified by the 2-back > 0-back task contrast on a fractal n-back working memory task, which reliably recruits the fronto-parietal network and distributed executive system (Ragland et al., 2002; Satterthwaite et al., 2013). As before, we calculated the control energy cost for each participant to transition from the baseline state to this new target state, and evaluated the associations of control energy and age. Consistent with our main results, the control energy cost for reaching this activation-based target state using real network data was significantly lower than that from a null network (see new Figure 2—figure supplement 4B). Similarly, control energy cost remained highest in frontoparietal system (Figure 2—figure supplement 4C). Critically, both the whole-brain control energy cost (Z = -7.59, P = 3.26 × 10^-14^, Partial r = -0.25, CI = [-0.30, -0.18]; Figure 2—figure supplement 4d) and control energy in the fronto-parietal system (Z = -5.26, PFDR = 2.92 × 10^-7^, Partial r = -0.17, CI = [-0.23, -0.11]; Figure 2—figure supplement 4E) both significantly declined with age. Nodal-level analyses provided convergent results, with control energy in fronto-parietal system nodes decreasing with development (Figure 2—figure supplement 4F). The Materials and methods section and Results section have been revised to reflect these new analyses:

Materials and methods section:

“Finally, we evaluated an alternative, biologically recorded target state that was defined using the activation pattern from a working memory task that reliably recruits the fronto-parietal network and executive system. […] Furthermore, we calculated the developmental association between control energy and age at multiple scales, including the whole brain, each cognitive system, and for each network node (with covariates as prior).”

Results section:

“Finally, in our main analyses, we specified the target state as regions within the fronto-parietal system, with each region having a magnitude of 1. […] Nodal analyses provided convergent results, revealing that the control energy in nodes within the fronto-parietal system significantly declined with age (Figure 2—figure supplement 4F).”

6) Equation 2: "S is 0-1 diagonal matrix of size N x N that selects only the nodes that we wish to control. Here, we only constrain the activity of the fronto-parietal system." – Can the authors clarify this statement? Is S the identity matrix? We thought that all nodes are targeted since the target state comprises 1 for the frontoparietal nodes and 0 for other nodes. But the authors now seem to imply they only constrain the activity of the frontoparietal nodes.

We are happy to clarify. The diagonal matrix *S* selects the nodes that will have a state penalty in the cost function. In our analyses, the diagonal matrix *S* was not an identity matrix, and only nodes in the fronto-parietal system had a magnitude of 1. This means there was a penalty associated with large state deviations only for the frontoparietal system. Our work aimed to examine how structural networks facilitate the selective activation of the fronto-parietal system. Therefore, we specifically care about the energy required to activate the fronto-parietal system without large deviations from 1 (with 1 being the target state value for frontoparietal regions). This choice also makes our results robust to other choices of state values, as evinced by the fact that both the initial and the target state values of these off-target regions did not impact the calculation of control energy. In the revised manuscript, we have clarified this point in the revised Results section:

“Third, it should be noted that we only constrained the state of regions in the fronto-parietal system. Therefore, the distance travelled by these off-target regions outside the fronto-parietal system were not included in our cost function for calculating optimal control energy. This choice also serves to ensure that our calculation of control energy is largely robust to both the initial and target states of other regions.”

While our choice of *S* has clear theoretical motivation, the reviewer is correct that it is a still a parameter whose impact on the results could be explored. To demonstrate the robustness of our results to our choice of *S*, in the revised manuscript we now include a sensitivity analysis that sets *S* as the identity matrix and constrains the activation of all nodes. Results showed there was a high correlation (r = 0.94, p < 2 × 10^-16^) between the whole-brain control energy cost when constraining the fronto-parietal system and the whole-brain control energy cost when constraining the whole brain (Figure 2—figure supplement 2D). This new result has been included in the revised manuscript:

“To demonstrate the robustness of our results to our definition of the matrix S, we calculated the control energy cost using the same initial and target states as in the main analyses but constraining the whole brain. Results showed that there is a high correlation (r = 0.94, p < 2 × 10^-16^) between the whole-brain control energy cost when constraining the whole brain and that when constraining the fronto-parietal system only (Figure 2—figure supplement 2D).”

7) "When model weights were examined at the level of individual network nodes, the regions that most contributed to the prediction of brain maturity aligned with univariate analyses, and included the dorsolateral and ventrolateral prefrontal cortex, the cingulate cortex, superior parietal cortex, and lateral temporal cortex" – It is well-known that model weights should not be interpreted without filtering (Haufe et al., 2014). Given the authors are using ridge regression, Haufe's approach is perfect for this situation.

We appreciate this excellent suggestion. In the revised manuscript, we used the suggested method (Haufe et al., 2014) to interpret the weights of the ridge regression model. Specifically, we multiplied the model weight vector *w* by the data covariance matrix ∑_X_ (Haufe et al., 2014; Waskom and Wagner, 2017), which was formulized as *a* = ∑_X_ • *w*. The transformed weight vector *a* was a distributed pattern quantifying the contribution of each brain region in multivariate ridge prediction. Results were consistent with mass-univariate analysis, which demonstrated that the brain regions contributing the most to the prediction included the dorsolateral and ventrolateral prefrontal cortex, the cingulate cortex, superior parietal cortex, and lateral temporal cortex (Figure 3B). We have described this new approach in the revised Materials and methods section:

“Interpreting the model: In a linear prediction model such as ridge regression, one weight/regression coefficient was assigned for each feature/brain region. […] The absolute value of the transformed weight represents the importance of the corresponding feature in a prediction (Haufe et al., 2014; Mourao-Miranda et al., 2005).”

We have also updated the Results section:

“We further examined model weights at the level of individual network nodes. The regions that contributed the most to the prediction of brain maturity aligned with mass-univariate analyses, and included the dorsolateral and ventrolateral prefrontal cortex, the cingulate cortex, superior parietal cortex, and lateral temporal cortex (Figure 3B).”

8) The authors need to provide more details about how they controlled for linear and nonlinear effects of age in their analysis (e.g., correlation between control energy and executive performance). The authors mentioned generalized additive models and penalized splines, but the details were nowhere close to being sufficient.

We are happy to elaborate. We used the following generalized additive model (GAM) to estimate the associations between control energy cost and executive performance.

Energy = Executive Performance + spline(Age) + Sex + Handedness + Motion + TBV + Network Strength + intercept.

This modeling approach fits penalized splines to flexibly model nonlinearities in the relationship between age and measures of interest, without being bound to specific linear or polynomial functions. The spline age term estimates a nonparametric smooth function for age-related associations with control energy, which can include linear or nonlinear effects depending on the structure of the data. Restricted maximum likelihood is used to penalize non-linearity in order to avoid over-fitting (Wood, 2004). In our data, the association between control energy and age was linear for the fronto-parietal system (Figure 2C), but we observed a non-linear association in the limbic system (Figure 2 —figure supplement 1D). Therefore, in the above model, GAMs controlled for both linear or non-linear age effects as dictated by the specific data. We have clarified these details in revised Materials and methods section:

“For the associations between control energy and executive performance, the GAM model was:

Energy = Executive Performance + spline(Age) + Sex + Handedness + Motion + TBV + Network Strength + intercept.

We used the gam command in the R package ‘mgcv’ to implement the model. The spline term estimates a nonparametric smooth function for age-related differences in control energy, which can include linear or nonlinear effects depending on the structure of the data. Restricted maximum likelihood is used to penalize non-linearity in order to prevent overfitting (Wood, 2004).”

9) The authors need to provide more details about the mediation analysis (e.g., assumptions). How do we interpret the betas in Figure 4C and D? Total effect is 0.67, while mediation effect is 0.03 and direct effect is 0.64. Doesn't this mean that the analysis is suggesting that the indirect effect is quite small relative to the direct effect? If so, this should be discussed.

We appreciate this comment, and we have added further details clarifying the mediation analysis in the revised Materials and methods section:

“Furthermore, for regions that displayed the associations between control energy and both age and cognition, we evaluated whether regional control energy might mediate the relationship between age and executive function. […] A bootstrap analysis (i.e., resampled 10,000 times) was implemented to estimate the confidence intervals for the indirect effect.”

We agree with the reviewer that the indirect effect was quite small relative to the direct effect. As suggested, in the revised manuscript, we now emphasize the small indirect effect size (Discussion section):

“Moreover, the decline of control energy of the bilateral middle cingulate cortex partially mediated the observed improvement of executive function with age. This result is consistent with prior literature suggesting that the optimization of the structural network is associated with better executive function (Baum et al., 2017; Wen et al., 2011), as the decline of control energy cost reflects the optimization of the structural brain network (Kim et al., 2018). However, it should be noted that this mediation effect was small, as the direct effect was much larger than the indirect effect.”

10) The age prediction analysis requires motivation and better integration with the rest of the paper. It is likely that simpler features such as the tractography connectivity strengths or basic network properties would also yield good predictive models of age. Therefore, the reason for using a more complex feature space is unclear, unless it can be demonstrated that control energy can outperform the accuracy of more basic features, or reveals a specific mechanism that characterizes network development.

We are happy to clarify this point. Multivariate pattern analyses complement mass-univariate approaches, as mass-univariate analyses investigates each brain region in isolation and multivariate pattern analyses are sensitive to the spatially distributed pattern of features (Davatzikos, 2004; Haynes, 2015; Haynes and Rees, 2006; Norman et al., 2006). We included a multivariate pattern analysis to explore whether the results were consistent with our mass-univariate analyses, and also to examine the total predictive power of the complex pattern of control energy. It is important to note that we do not emphasize the relative predictive power of these control energy features compared to simpler features; this analysis is intended to provide an integrated view of the high-dimensional data. Results were consistent with the mass-univariate analysis, which underscores the robustness of these findings to the methodological approach. In the revised manuscript, we have clarified the motivation for the multivariate pattern analysis (Results section):

“Having established that the control energy required to reach the fronto-parietal activation state changes with age on a regional and system-level basis using mass-univariate analysis, we next evaluated the developmental changes of control energy using multivariate pattern analysis. […] Specifically, we applied ridge regression with nested two-fold cross validation (2F-CV, see Figure 3—figure supplement 1) to identify an individual participant’s age in an unbiased fashion using the multivariate pattern of regional-level control energy.

11) The relation identified between control energy and age appears to be linear based on Figure 2 and thus the reasons for fitting penalized splines to characterize nonlinear associations could be better described. Are the lines shown in Figure 2 representative of the splines?

We thank the reviewer for their comment. Because prior studies demonstrated that the developmental changes of brain structure and functions could be either linear (Hagmann et al., 2010; Wierenga et al., 2016) or non-linear (Grayson and Fair, 2017; Mills et al., 2016; Vandekar et al., 2015), we used generalized additive models (GAMs) to fit penalized splines, which characterized both linear and nonlinear associations between age and control energy cost. The GAM formula used to model the effect of development on energy was:

Energy = spline(Age) + Sex + Handedness + Motion + TBV + Network Strength + intercept.

We used the gam command in the R package ‘mgcv’ to implement the model. Penalized splines allow the model to capture both linear and non-linear relationships with age, while penalizing overfitting using relative maximum likelihood. For example, while the plot shown in Figure 2A suggested the relationship between whole-brain average control energy and age was linear. However, associations between age and average control energy in limbic system were non-linear (see new Figure 2—figure supplement 1D).

In revised manuscript, we now elaborate regarding the motivation for using penalized splines (Results section):

“Prior studies have demonstrated that the developmental changes of both brain structure and function could be either linear (Hagmann et al., 2010; Wierenga et al., 2016) or non-linear (Grayson and Fair, 2017; Mills et al., 2016; Vandekar et al., 2015). Therefore, we used generalized additive models (GAM) with penalized splines, which allowed us to rigorously characterize both linear and nonlinear effects while avoiding over-fitting.”

We also added the details about the implementation of GAM in the revised Materials and methods section.

“For developmental effect of control energy, the GAM model was:

Energy = spline(Age) + Sex + Handedness + Motion + TBV + Network Strength + intercept.

For the associations between control energy and executive performance, the GAM model was:

Energy = Executive Performance + spline(Age) + Sex + Handedness + Motion + TBV + Network Strength + intercept.

We used the gam command in the R package ‘mgcv’ to implement the model. The spline term estimates a nonparametric smooth function for age-related differences in control energy, which can include linear or nonlinear effects depending on the structure of the data. Restricted maximum likelihood is used to penalize non-linearity in order to prevent overfitting (Wood, 2004).”

The claims regarding the specificity of the FP network require clarification. The null model corresponding to the rewired graph does not appear to have been evaluated for the other canonical networks. To establish specificity of the FP, the null model should also be considered for the other networks. Based on this global null model alone, we would suggest that it is hard to claim that the brain is explicitly wired to optimize transition cost to the FP activation state. Wiring organization is also likely to contribute to other cognitive functions as well.

This is a valuable comment. To address the reviewer’s point, we also evaluated the null model for a motor target state. Results indicated that the mean whole brain energetic cost of the null networks was also significantly higher (p < 2 × 2^-16^) than that of the empirical networks (see new Figure 2—figure supplement 2E left). As the reviewer suggested, this suggests that the lower energetic cost of the fronto-parietal system compared to null networks was not unique but was present in other systems as well. We have added this result in the revised manuscript.

Materials and methods section:

“Third, we assessed whether the structural network also contributed to other cognitive functions as well by comparing the control energy cost required to reach a motor activation state for real networks and null networks.”

Results section:

“Fourth, we assessed whether the structural network optimized the transition to an a priori motor system activation target (Figure 1—figure supplement 2) (Yeo et al., 2011). Results indicated that the mean whole brain energetic cost of the null networks was significantly higher (p < 2 × 10^-16^) than that of the empirical networks (Figure 2—figure supplement 2E left), suggesting that the lower energetic cost of the fronto-parietal system was not unique, but was present in other systems as well.”

Based on these results, we have removed statements regarding specificity of the fronto-parietal network in the Materials and methods section, Results section and Discussion section.

However, it should be noted that when developmental associations with control energy cost were evaluated for the motor system, we found that the whole-brain control energy required to transition to the motor system activation did not significantly change over the age range studied (Z = 1.48, P = 0.14, Partial r = 0.05, CI = [-0.02 0.11]; Figure 2—figure supplement 2E right). This result is consistent with extensive literature (Andersen, 2003; Gogtay et al., 2004) demonstrating more protracted neurodevelopment of executive versus motor systems.

12) In addition, it would be interesting to know whether the node-level measures are related to degree and/or strength. The authors use the two statistics as covariates in the predictive model, but do not show whether they are related to control energy directly.

This is a good point. As suggested, we tested the association between the total network strength and whole-brain control energy, while controlling for a spline of age, sex, handedness, motion, and total brain volume. Results suggested the association between total network strength and control energy was not significant (Z = 0.79, P = 0.43, Partial r = 0.03, CI = [-0.04, 0.09]). It should be noted that we have scaled the network matrix and we have controlled for the total network strength of this scaled network. It should be noted that the energy required for a state transition depends upon a system’s response to an energetic perturbation. The simplest such perturbation engenders an impulse response (Karrer et al., 2020). In Srivastava et al., (2020), we show that the impulse response of the system is formally related to the network communicability, suggesting that paths of all lengths – not just direct connections as quantified in degree or strength – contribute to control energy.